# Surface melt on the Shackleton Ice Shelf, East Antarctica (2003–2021)

Dominic Saunderson[1], Andrew Mackintosh[1], Felicity McCormack[1], Richard S. Jones[1], and Ghislain Picard[2, 3]

[1]Securing Antarctica's Environmental Future, School of Earth, Atmosphere and Environment, Monash University, Clayton, VIC 3800, Australia
[2]Univ. Grenoble Alpes, CNRS, Institut des Géosciences de l'Environnement (IGE), UMR 5001, Grenoble, France
[3]Geological Survey of Denmark and Greenland (GEUS), 1350 Copenhagen, Denmark

**Correspondence:** Dominic Saunderson (dominic.saunderson@monash.edu)

**Abstract.** Melt on the surface of Antarctic ice shelves can potentially lead to their disintegration, accelerating the flow of grounded ice to the ocean and raising global sea levels. However, the current understanding of the processes driving surface melt is incomplete, increasing uncertainty in predictions of ice shelf stability and thus of Antarctica's contribution to sea-level rise. Previous studies of surface melt in Antarctica have usually focused on either a process-level understanding of melt through

energy-balance investigations, or used metrics such as the annual number of melt days to quantify spatiotemporal variability in satellite observations of surface melt. Here, we help bridge the gap between work at these two scales. Using daily passive microwave observations from the AMSR-E and AMSR-2 sensors, and the machine learning approach of a self-organising map, we identify nine representative spatial distributions ("patterns") of surface melt on the Shackleton Ice Shelf in East Antarctica from 2002/03-—2020/21. Combined with output from the RACMO2.3p3 regional climate model and surface topography

from the REMA digital elevation model, our results point to a significant role for surface air temperatures in controlling the interannual variability of summer melt, and also reveal the influence of localised controls on melt. In particular, prolonged melt along the grounding line shows the importance of katabatic winds and surface albedo. Our approach highlights the necessity of understanding both local and large-scale controls on surface melt, and demonstrates that self-organising maps can be used to investigate the variability of surface melt on Antarctic ice shelves.

**1   Introduction**

Much of the uncertainty in projections of sea-level rise stems from an incomplete knowledge of the processes causing mass loss from the Antarctic Ice Sheet (Bamber et al., 2019; Oppenheimer et al., 2019; Robel et al., 2019). This uncertainty is closely linked to the stability of the continent's ice shelves, which are floating extensions of the ice sheet, and surround ~ 75 % of the grounded ice (Bindschadler et al., 2011). Ice shelves are key to potential non-linear responses to climate change (Weertman,

1974; Pattyn and Morlighem, 2020; Sun et al., 2020), and can control the ice sheet mass balance by buttressing the flow of grounded ice to the ocean (Dupont and Alley, 2005; Fürst et al., 2016; Haseloff and Sergienko, 2018). If the buttressing force

of the floating ice is reduced (e.g. via thinning, calving, and/or topographic unpinning), or the shelf collapses, the upstream ice can accelerate, leading to increased mass loss (Rott et al., 2002; Scambos et al., 2004; Rignot et al., 2004; Konrad et al., 2018).

One of the most notable examples of ice shelf collapse occurred in 2002, when the Larsen B Ice Shelf disintegrated in six weeks (Rack and Rott, 2004), following intense surface melt (Sergienko and MacAyeal, 2005; van den Broeke, 2005) and extensive meltwater ponding across the shelf surface (Leeson et al., 2020). Such ponding occurs when the firn layer becomes saturated, depleting the firn air content of the shelf (Kuipers Munneke et al., 2014; Holland et al., 2011; Luckman et al., 2014), and allowing liquid water to collect in depressions on the ice shelf surface (Arthur et al., 2020a). If the stress of the ponded meltwater overcomes the ice overburden pressure, crevasses can penetrate through the shelf (i.e. hydrofracture) (Weertman, 1973; van der Veen, 1998; Scambos et al., 2000), and potentially initiate cascading effects that lead to the rapid loss of large areas of ice (MacAyeal et al., 2003; Banwell et al., 2013; Robel and Banwell, 2019).

Broadly speaking, previous research has investigated the occurrence of surface melt in Antarctica in one of two ways. Firstly, surface energy balance (SEB) studies have allowed process-level insights into melt by identifying and explaining the responsible energy fluxes. SEB studies require the availability of in-situ weather observations (van den Broeke et al., 2010; Nicolas et al., 2017) or modelling output (King et al., 2017; Scott et al., 2019), and such studies are therefore often limited to examining individual melt events (i.e. over several days) (Zou et al., 2019; Ghiz et al., 2021) or a few melt seasons (Elvidge et al., 2020; Turton et al., 2020), though some longer, multiannual records do exist (Jakobs et al., 2020).

Secondly, previous studies describe quantitatively the occurrence and extent of melt in Antarctica using a series of melt metrics calculated from satellite observations. Typical metrics include the melt onset and freeze-up dates each summer, the total number of melt days, and the cumulative melting surface (e.g. Zwally and Fiegles, 1994; Torinesi et al., 2003). These metrics are often reported at a regional (e.g. Antarctic Peninsula, Wilkes Land) or continental scale, and usually show large interannual variability, with only short-term or insignificant trends (e.g. Liu et al., 2006; Picard et al., 2007; Tedesco et al., 2007; Trusel et al., 2012; Zheng et al., 2018; Johnson et al., 2021). Studies using melt metrics that have focused on individual shelves have largely been restricted to the Antarctica Peninsula (e.g. Bevan et al., 2018, 2020; Banwell et al., 2021), with only a couple in East Antarctica (Zhou et al., 2019; Zheng and Zhou, 2020; Dell et al., 2021).

In this paper, we investigate the spatial variability of surface melt on the Shackleton Ice Shelf in East Antarctica. This shelf experiences the most intense melt in Antarctica beyond the Antarctic Peninsula (Trusel et al., 2013), and hosts supraglacial lakes each summer (Arthur et al., 2020b). We use a machine learning approach to assess the inter- and intra-annual variability in the location of satellite-observed surface melt, and compare our results with surface topography and patterns of climate variables such as surface air temperatures and albedo to identify the influence of localised controls on the occurrence of surface melt. Our approach therefore goes beyond the metrics used in previous remote sensing studies and begins to bridge the gap between spatiotemporal descriptions of melt variability derived from satellite observations, and the process-level understanding of melt from SEB studies that are more detailed but limited in time and space.

The paper is structured as follows. In Sect. 2, we provide an overview of the Shackleton Ice Shelf and the passive microwave datasets. In Sect. 3, we describe the self-organising map methodology. In Sect. 4, we present the results, which we then discuss in Sect. 5; particular attention is given to understanding the results in relation to the local geographic setting of the shelf

(e.g. surface topography, albedo, and winds) and its role in controlling surface melt. Finally, in Sect. 6, we briefly conclude the work.

## 2 Study region and datasets

### 2.1 Shackleton Ice Shelf

The Shackleton Ice Shelf (~ 66º S, ~ 100º E) is the northernmost major ice shelf in Antarctica (Fig. 1), and covers an area of ~27 000 $km^2$ (Mouginot et al., 2017; Rignot et al., 2013). The shelf extends for > 250 km along the Queen Mary Land coast, and is fed by several fast-flowing glaciers, between which lie areas of slow-moving ice constrained by a series of islands, ice ridges and ice rumples (Stephenson and Zwally, 1989). Only ~ 30 % of the shelf is considered to be passive ice that does not provide any buttressing force (Fürst et al., 2016).

The largest of the outlet glaciers is the Denman Glacier, which contains a sea-level rise equivalent of 1.5 m and sits on a retrograde slope connected to the large Aurora Subglacial Basin (Morlighem et al., 2020). The Denman Glacier is estimated to have lost ~ 190 Gt of ice since 1979 (~ 0.5 mm of sea-level rise; Rignot et al., 2019), and has accelerated over both its grounded and floating portions (1972-—2017; Miles et al., 2021), with its grounding line having retreated nearly 5.5 km between 1996 and 2018 (Brancato et al., 2020; Konrad et al., 2018). Understanding the response of the Denman Glacier and wider Shackleton system to climate change and variability is therefore an important area of research.

### 2.2 Satellite datasets

We use daily passive microwave observations of surface brightness temperature ($T_B$) at 19 GHz from two successive versions of the Advanced Microwave Scanning Radiometer (AMSR), called AMSR-E (May 2002–October 2011) and AMSR-2 (May 2012–present). The 19 GHz observations have an underlying footprint of ~ 14–16 km x 22–27 km, but are pre-processed to a regular 12.5 km southern stereographic polar grid using the drop-in-the-bucket method with daily averaged $T_B$ (Meier et al., 2018). The overpass time, which is important for climatic studies of surface melt (Picard and Fily, 2006), has remained approximately constant through the sensors' lifespans, with the equatorial crossing time of the ascending pass being ~13:30 each day (REMSS, 2022).

To assess whether the time-series can be extended back in time, we further utilise observations from the Special Sensor Microwave Imager (SSM/I) and the Special Sensor Microwave Imager Sounder (SSMIS) sensors: F13 (May 1995–December 2007), F17 (December 2006–present), and F18 (January 2017–present). These observations, collectively hereafter *SSMIS*, were pre-processed in the same way as the AMSR datasets, but gridded at 25 km due to their coarser underlying footprint (~ 70 km x 45 km at 19 GHz) (Meier et al., 2021). Overpass times vary between the three SSMIS sensors, with their ascending passes observing the Shackleton Ice Shelf between ~ 16:30 and ~ 20:00 local solar time each day.

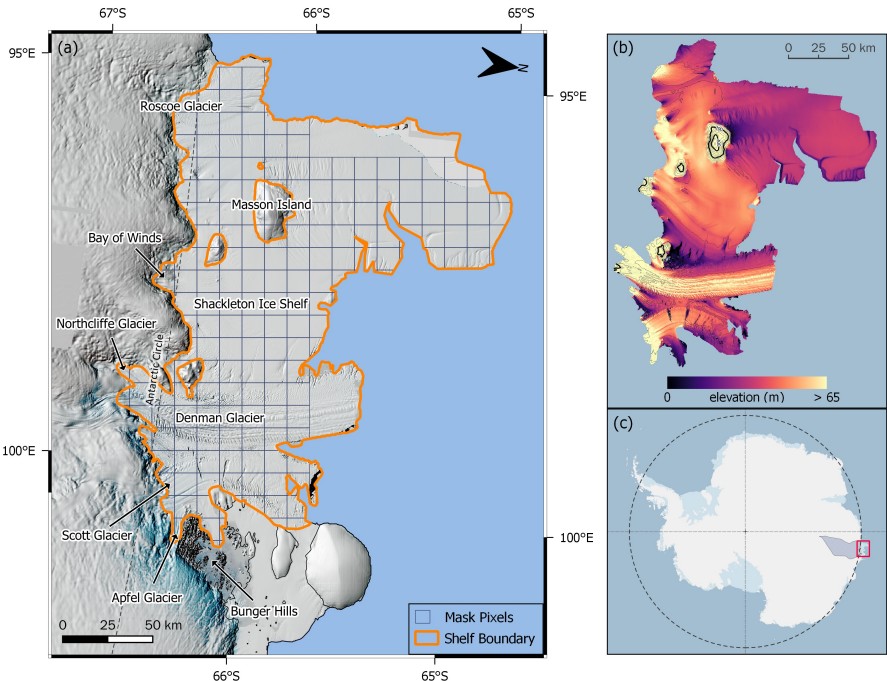

**Figure 1.** (a) Geographic setting of the Shackleton Ice Shelf. The background image shows LIMA imagery (2000–2003) (Bindschadler et al., 2008) overlain on REMA topography (Howat et al., 2019); the shelf boundary is taken from the MEaSURES dataset (Mouginot et al., 2017). (b) Surface elevation of the Shackleton Ice Shelf using REMA topography; contour lines are shown at elevations of 10 m (light grey), 40 m (black), and 200 m (thick black). (c) Location of the Shackleton Ice Shelf in Antarctica; the basin which drains through the shelf is also shown. Figure created using Quantarctica (Matsuoka et al., 2021).

## 2.3 Melt detection

Brightness temperature can be understood as the product of a surface's physical temperature and its emissivity. The emissivity of dry snow (0.65–0.8 at 19 GHz in horizontal polarisation) is much lower than that of wet snow (0.95), meaning that large increases in $T_B$ (e.g. 30 K) can be explained by the presence of liquid water rather than an increase in the physical temperature of the surface (Zwally and Fiegles, 1994).

Using the horizontally-polarised observations at 19 GHz, which are particularly sensitive to changes in emissivity, we process the data according to the algorithm used in Picard and Fily (2006). This algorithm uses a threshold approach to detect melt, with the threshold calculated for each pixel individually and redefined each summer. The threshold is calculated as the sum of the mean and 2.5 times the standard deviation of $T_B$ observations for dry snow each year (1st April—31st March). Dry snow is defined recursively, iteratively removing any observations identified as wet snow and recalculating the melt threshold using only the remaining observations, until no further observations need to be removed; one or two iterations are sufficient to reach convergence. A full explanation can be found in Torinesi et al. (2003).

Using an adaptive threshold accounts for changes in the snowpack (e.g. melt cycles, grain metamorphism, precipitation; Kunz and Long, 2006) between years and regions, and is thus more robust for melt detection. The result of the algorithm is a binary distinction, for each pixel, for each day, of whether liquid water is present at the surface or not, respectively understood henceforth as representing "melt" and "no-melt" conditions.

## 2.4 Shelf mask

To define the boundary of the Shackleton Ice Shelf, we make two small modifications to the shelf mask from the MEaSUREs dataset (Rignot et al., 2013; Mouginot et al., 2017). Firstly, we exclude the front ~ 25 km of the Denman Glacier tongue, where the pixels becomes progressively more ice-covered each summer owing to the glacier's high flow velocity (Miles et al., 2021). Secondly, we exclude the westernmost edge of the shelf, which is often bordered by a polynya (Nihashi and Ohshima, 2015). Manual inspection of the underlying $T_B$ data indicates that the pixels along the western edge may have been contaminated by the inclusion of sea ice and open ocean in the sensor footprint, and are therefore not suitable for use with our algorithm, which is only designed to differentiate between wet and dry snow. The final shelf mask is shown in Fig. 1a; any calculations regarding melt area are weighted to discount non-shelf area within the pixels. To compare between the two datasets, we resample the SSMIS binary melt data (25 km pixels) to the resolution of the AMSR data (12.5 km) using a nearest neighbour approach.

## 2.5 Melt metrics

For each pixel within the shelf mask, we calculate four annual melt metrics used in the literature: the melt onset and freeze-up dates each summer; the number of days between these two dates (melt season "length"); and the number of days during the melt season when melt is observed (melt season "duration"). We further calculate what we term the melt season "fraction", which is simply the duration of the melt season divided by its length, and is therefore a measure of how consistently a pixel experiences melt throughout a summer.

We calculate the cumulative melting surface (CMS), measured in day km$^2$, by summing the daily melt extents each summer (Torinesi et al., 2003; Picard and Fily, 2006). It is also known as the Melt Index (Zwally and Fiegles, 1994; Trusel et al., 2012).

Finally, to assess when different patterns of melt occur within their respective melt season, we assign each day in a summer a melt season "context". The context is defined as how many days into a melt season a melt observation occurs, expressed as a percentage of the respective melt season's full length. This metric was used to test the hypothesis that all melt seasons initiate, develop, and terminate in the same way regardless of their length or specific calendar dates.

## 2.6 Climate variables

We use monthly-averaged climate variables from the RACMO2.3p3 regional climate model (van Dalum et al., 2021, 2022), which is the latest version of the RACMO model that has been used extensively in Antarctica (e.g. Lenaerts et al., 2012; van Wessem et al., 2014a, 2018). RACMO2.3p3 includes updates to the albedo scheme and multilayer firn module, as well as

allowing subsurface penetration of shortwave radiation, which can be important for melt in Antarctica (Liston and Winther, 2005; Liston et al., 1999). The model is forced at its lateral boundaries by ERA5 reanalysis data (Hersbach et al., 2020).

## 3  Self-organising maps

A self-organising map (SOM) is a machine learning approach that can simplify multi-dimensional datasets (Kohonen, 1990, 2001). SOMs have been widely used in many fields, including climate science (e.g. Hewitson and Crane, 2002; Sheridan and Lee, 2011; Cassano et al., 2016; Gibson et al., 2017; Udy et al., 2021), but have not been used to investigate surface melt in Antarctica before.

A SOM algorithm produces a set of outputs that represent important variability in the input data. We refer to these outputs as *patterns*, because, for our dataset, each output represents a typical spatial distribution (i.e. "pattern") of melting and non-melting pixels across the shelf, taken at a daily resolution.

SOMs work by comparing each input (here a daily melt observation) against a set of self-adaptive reference nodes, which ultimately become the output melt patterns. The similarity between an input and each of the nodes is judged by a user-defined similarity measure (Sect. 3.1), and the input assigned to the least dissimilar node. In turn, the node slightly adapts to better represent its newest member; neighbouring nodes also adapt to ensure the data self-organise. This full process is iterated through multiple times, with each input repeatedly fed into the algorithm until it converges to a solution (see Sect. 1 in the Supplement for further explanation). The self-organisation ensures that the final melt patterns represent a continuum of the input data. Being able to produce a continuum of melt patterns is one of the main reasons why we use a SOM approach over, for example, an empirical orthogonal function approach, which requires orthogonal outputs and therefore often loses physical meaning after the first or second leading modes.

### 3.1  Implementation with binary surface melt data

We apply the SOM approach using the *kohonen* package (Wehrens and Buydens, 2007; Wehrens and Kruisselbrink, 2018) in R (R Core Team, 2021). Daily maps of binary melt status in the AMSR datasets are flattened to 1D vectors and combined into a matrix in which each row represents a single day, and each column references a specific pixel within the shelf mask. All days between the onset and freeze-up dates of each summer are included. Owing to the binary nature of our input data, we use the "Tanimoto" similarity measure. Within the kohonen package, this setting represents the Hamming Distance divided by the total length of the input vector; conceptually, this is the fraction of times in which the input disagrees with the reference pattern.

Implementation of the SOM algorithm requires the definition of multiple initiating parameters, the most crucial here being how many output patterns are required. We run a series of sensitivity experiments for each of the parameters (Fig. S4–S5) and find that using nine patterns is the most appropriate for our purposes (Sect. 3.3). Boxplots showing how well each SOM pattern represents its respective members are shown in the supplement, along with examples of visual comparisons (Fig. S2–S3). Together, these figures show that the SOM patterns are able to capture the dominant features of spatial variability and simplify the dataset for further analysis.

## 3.2 Interpretation of the self-organising map output

As stated above, each of the nine output patterns from the SOM algorithm (Fig. 2) represent a typical distribution of surface melt across the Shackleton Ice Shelf on a daily basis. These output patterns should be understood as being representative models of melt and non-melt pixels: the value of each pixel indicates the likelihood of melt occurring in the pixel for a given pattern. For example, a pixel value of 0.9 in pattern X indicates that the pixel melts on 90% of the days when the observed melt distribution is best described by pattern X (see Sect. S1 and Fig. S1 for further details). Pixel values close to 0 or 1 therefore show pixels with more consistent melt behaviour on days assigned to the pattern, and can be interpreted as being more important to the definition of the pattern. In contrast, values closer to 0.5 show pixels which are nearly equally likely to melt or remain frozen within the pattern.

## 3.3 Limitations of self-organising maps

A SOM algorithm does not produce a single, objectively "correct" solution. The algorithm is designed to help identify patterns and form groups within the input data, and thus whether the output is "correct" depends on the question being asked and the data being used.

Inspecting the output from our sensitivity tests shows that the same spatial patterns appear repeatedly, regardless of the SOM parameters used (Fig. S4). Although the outputs are not identical across the tests, it must be remembered that the underlying AMSR data have already been pre-processed and gridded to a higher resolution, and are thus only suitable for identifying broader spatial patterns rather than highlighting specific differences in individual pixels. This constraint also means that even though using more patterns may produce a slightly lower mean distance between the input data and the SOM patterns (Fig. S4d & S5), the additional patterns are near-replicas of the nine patterns shown in Fig. 2, and would be considered analogous when interpreting the results given the above constraints. Overall, the similarity of the outputs strengthens our confidence in the results, and our interpretations remain unchanged between sensitivity tests.

Were the current analysis to be repeated on an alternative shelf, or even on the Shackleton Ice Shelf using a different underlying dataset, there is no expectation that the same settings would produce the most optimal output for that particular case, and sensitivity tests for SOM parameters should be run again.

## 4 Results

The spatial variability of surface melt on the Shackleton Ice Shelf can be represented with nine characteristic melt patterns (Fig. 2). Here we describe each of the patterns, when they occur within a melt season (Fig. 3; Fig. S6), and common daily progressions between them (Fig. 4). We also look at their interannual variability (Fig. 5; Fig. S8). We discuss the causes of the different patterns in Sect. 5.

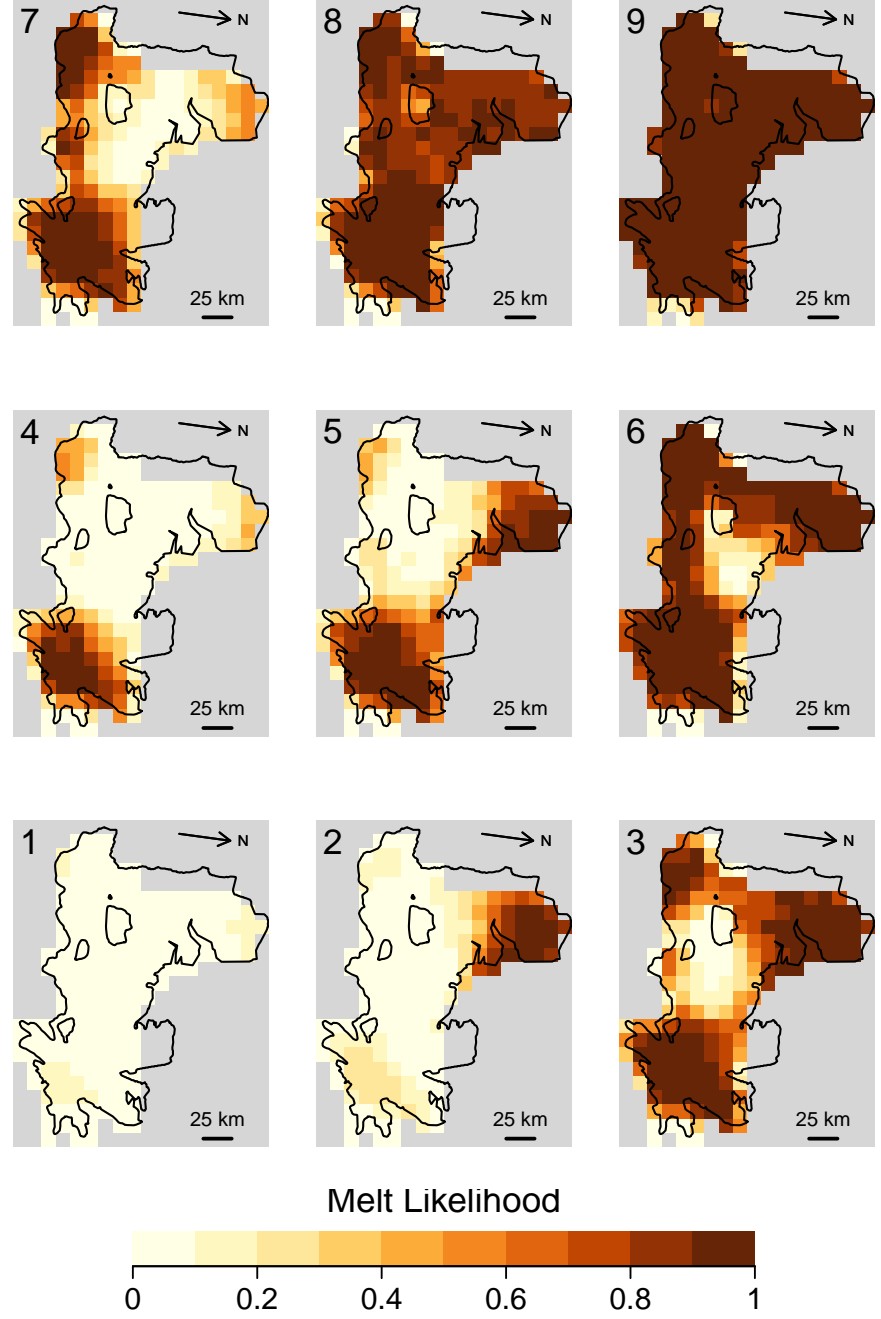

**Figure 2.** Result of the self-organising map, showing that nine representative melt patterns can describe the spatial variability of surface melt on the Shackleton Ice Shelf (2002/03–2020/21). Pixel values indicate how likely melt will be observed in a pixel on the days best described by the respective pattern; further explanation can be found in Sect. 3.2 and S1.

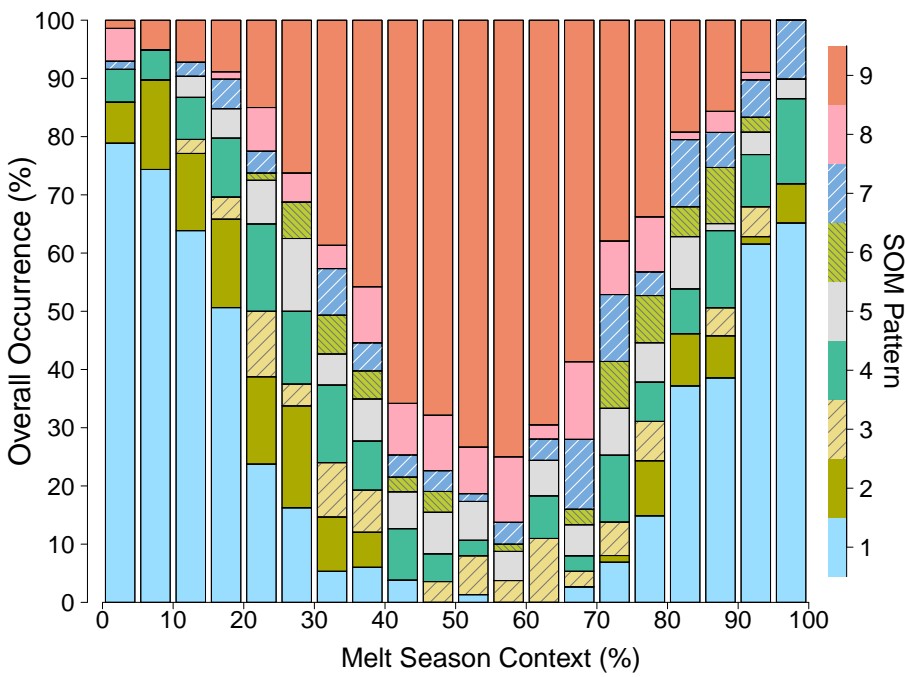

**Figure 3.** Timing of when the nine melt patterns occur within a melt season, plotted across all melt seasons. The x-axis indicates how far through the melt season a pattern occurs, expressed as a percentage of the respective season's full length. Y-axis values are stacked to indicate the relative occurrence of each pattern. The same data are plotted against calendar dates in Fig. S6.

## 4.1 Full shelf melt (Patterns 8 & 9)

Pattern 9 represents days when melt is observed across the full shelf (Fig. 2). It is the most common pattern overall, occurring on more than one-third of all melt season days across the 18 summers, and is observed for ~ 30 days each summer on average (Table 1). Pattern 8 is similar to pattern 9, but much less commonly observed and includes small intermittent dry areas without melt: on average, ~ 86 % (SD = 9.0 %) of the shelf experiences melt on pattern 8 days, compared to ~ 96.5 % (SD = 4.1 %) on pattern 9 days (Table 1; Fig. S2d). Both of these patterns can occur throughout the melt season (i.e. any time between the melt season onset and freeze-up dates), but are most prevalent mid-season (Fig. 3). Pattern 9 is also the most persistent pattern: ~ 90 % of all pattern 9 days are followed by another pattern 9 day (Fig. 4). In one instance, in 2019/20, pattern 9 persists for 44 straight days, and a similarly consistent sequence in 2005/06 is interrupted by only a single day of pattern 8. Interannually, the occurrence of patterns 8 and 9 are negatively correlated (r = -0.49; p = 0.04; Table S1), and only in three summers is pattern 8 observed more frequently than pattern 9 (2006/07–2008/09; Fig. S8).

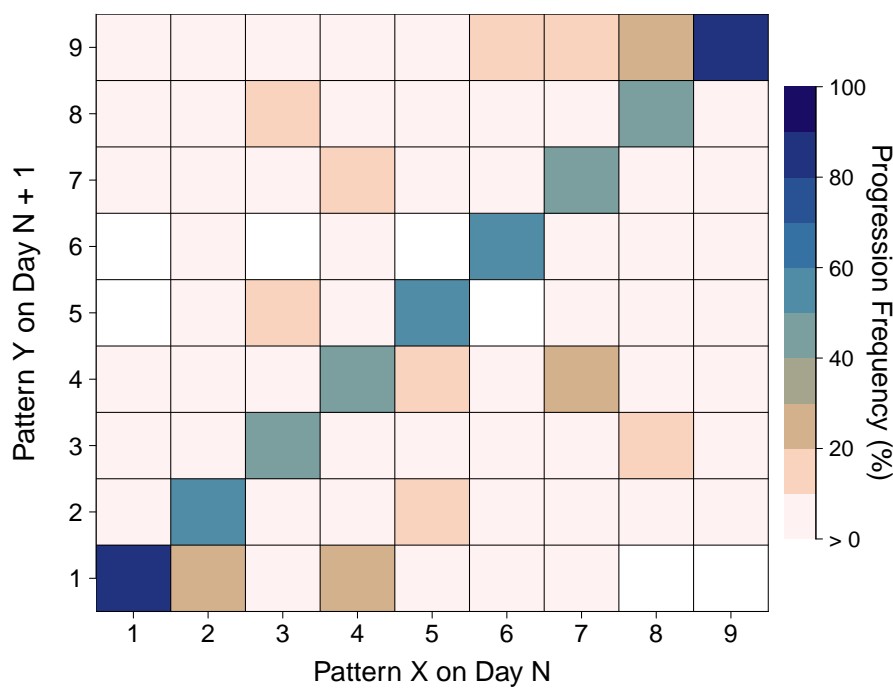

**Figure 4.** Day-to-day progressions between the nine melt patterns. The colours indicate how frequently a pattern on the x-axis develops into a pattern on the y-axis the following day, highlighting the strong tendency for any pattern to persist until at least the next day. White squares indicate that the progression is never observed.

## 4.2 Grounding line melt and dry shelf centre (Patterns 3, 6 & 7)

In patterns 3 and 6, melt is observed across much of the shelf, but the shelf centre remains dry in both patterns (Fig. 2). Melt along the grounding line is more extensive in pattern 6 than pattern 3, and the former has a slight tendency to develop into pattern 9 (Fig. 4), whereas the latter more commonly progresses to pattern 8, or recedes to pattern 5. Pattern 6 is also more prevalent in the last quarter of a melt season than earlier on (Fig. 3); pattern 3 is most frequently observed mid-season. However, care must be taken not to overinterpret these two patterns because they are infrequently observed overall (Table 1), and primarily occur only in a couple of summers each (Fig. 5; Fig. S8). It is therefore not clear whether these patterns are typical of the Shackleton Ice Shelf, or represent anomalous behaviour.

Pattern 7 also shows extensive melt along the grounding line and a dry shelf centre, but only very occasional melt in the north (Fig. 2). It has the highest variation between the melt observations that it best describes (Table 1), which is mainly realised in how far melt extends from the grounding line (Fig. S2c). Pattern 7 is infrequently observed in the first half of the melt season, but it becomes relatively common in the last third (Fig. 3). It often recedes to pattern 4 (Fig. 4), which in turn recedes to pattern 1: this progression (7 to 4 to 1) is a common way for a melt season to conclude.

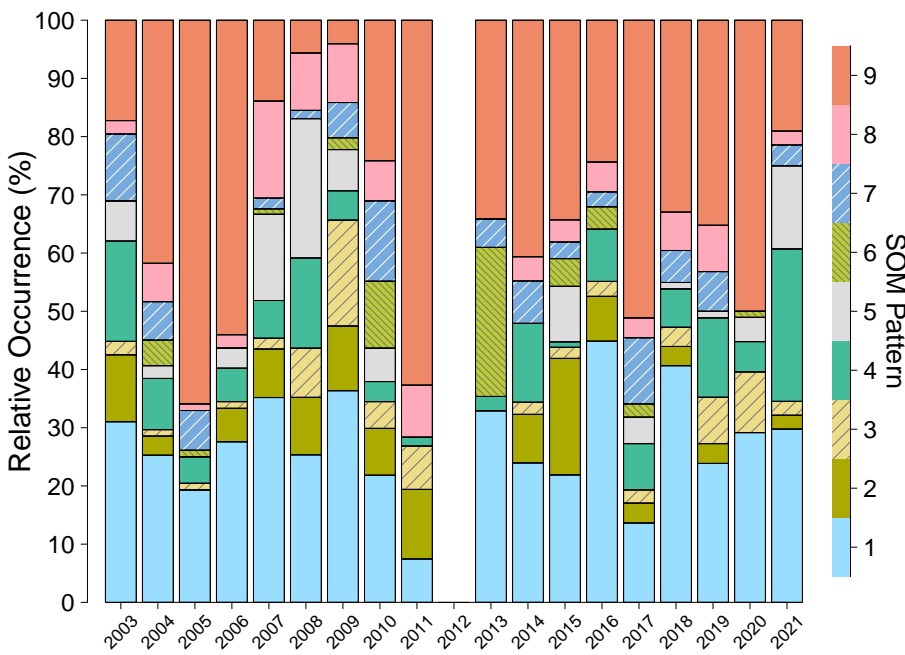

**Figure 5.** Relative occurrence of the nine melt patterns each summer. A value of 20 % on the y-axis indicates that 20 % of the days in a given melt season were best described by the given pattern. Years on the x-axis refer to the January date of the summer (i.e. 2016 is 2015/16). No AMSR sensor operated in 2011/12. See also Fig. S8.

### 4.3 Isolated melt zones (Patterns 2, 4 and 5)

In pattern 2, melt is restricted to the northernmost part of the shelf whilst the rest remains dry (Fig. 2). This pattern characterises
most of the significant melt that ever occurs before December in the summer (Fig. S6) and is common as the melt season develops (Fig. 3). It is then not seen at all during the middle third of the melt season, before reappearing again towards the end, albeit less frequently. Pattern 2 is more likely to recede back to a dry shelf (i.e. pattern 1) than develop into a more extensive melt pattern (Fig. 4).

Pattern 4 shows extensive melt in the south-east of the shelf, which is often accompanied by a small area of melt to the
220 south-west (Fig. 2). This pattern is a clear intermediate step in the shelf's seasonal melt evolution, observed as the melt season either intensifies or wanes, but very rarely at its peak (Fig. 3). On average, pattern 4 occurs on more than 6 days each summer, and is the third most common pattern overall after 9 and 1 (Table 1).

In pattern 5, extensive melt is observed to the north and the south-east of the shelf, along with sporadic melt in the south-west (Fig. 2). Pattern 5 therefore resembles a combination of patterns 2 and 4, which are the two patterns it most commonly

develops into (Fig. 4). Annually, the occurrence of pattern 5 is positively correlated with that of pattern 8 (r = 0.49; p = 0.04) and negatively correlated with that of pattern 9 (r = -0.61; p = 0.01; Fig. S8; Table S1).

## 4.4 Dry shelf (Pattern 1)

Pattern 1 represents days when the shelf is completely, or near-completely, frozen. It is the second most prevalent pattern overall (Table 1), and occurs towards the beginning or end of a melt season (Fig. 3). This pattern therefore represents either:
1) the slow onset or termination of a melt season, when melt is only sporadic and spatially confined to a few pixels; or 2) true non-melt days which occur between the melt season proper and distinct pre- or post-season melt events (see Fig. S7). In definitions of a persistent melt season onset (e.g. Tedesco et al., 2007), many of these days would not be considered as the melt season.

## 4.5 Interannual variability

We do not observe any statistically significant trends in the occurrence of any patterns on an interannual basis (Table 1; Fig. S8), nor any statistically significant covariation in the patterns (Table S1) other than in patterns 5, 8 and 9 as discussed above. Rather, the occurrence of the patterns displays strong interannual variability, particularly in absolute terms for pattern 9 (median absolute deviation (MAD) = 13 days). When normalised against the pattern's median occurrence, pattern 1 is the least variable pattern (coefficient of variation = 19 %). This value shows that the shelf consistently remains dry for a large proportion of days
between the first and last melt events each summer. We also observe the possibility of extreme values in a pattern's annual occurrence (Fig. 5; Fig. S8); for all patterns except pattern 9, the maximum annual occurrence of the pattern is larger than its median occurrence by at least three times its MAD; for patterns 3 and 6, the respective values are 16 and 20.

Together, the above observations show that certain melt patterns can be favoured within a melt season even if they are relatively uncommon overall, and suggests potentially significant differences in melt behaviour between summers. Examining
the relative occurrence of the nine patterns each summer (Fig. 5; Fig. S8) provides a snapshot of these differences. For example, in 2007/08 and 2008/09, minimums in the occurrence of pattern 9 days and maximums in the occurrence of patterns 5 and 3, respectively, show that the shelf centre remained unusually dry for most of these melt seasons. In contrast, in 2010/11, the low number of pattern 1 days and high proportion of pattern 9 days suggests that the melt season started and ended more abruptly than in other summers, and that melt occurred across the full shelf more consistently throughout. Future work can build on this
approach to identify and investigate such summers in further detail to gain a better understanding of the temporal variability of surface melt on Antarctic ice shelves.

**Table 1.** Summary statistics for the nine representative melt patterns. Frequency is the percentage of all melt season days on which the pattern occurs across all 18 summers. Mean extent is the average shelf area that melts for a pattern, with the standard deviation shown in brackets. Distance refers to the average Tanimoto distance between the SOM pattern and the observed melt distributions it represents. The coefficient of variation (CV) is defined as the median absolute deviation (MAD) divided by the median of the annual pattern counts (Liu et al., 2006). Trends are calculated with a Mann-Kendall linear trend test; no values are statistically significant. Bold values indicate correlations significant at $p < 0.05$. Summer temperatures are from RACMO2.3p3 for the area depicted in Fig. 7b. The cumulative melting surface (CMS) is defined as explained in Sect. 2.5.

| Pattern | Frequency (%) | Mean Extent (%) | Distance | Annual Daily Occurrence | | | | Correlation | |
| | | | | Median | MAD | CV(%) | Trend | Annual CMS | Mean DJ Temp. |
|---|---|---|---|---|---|---|---|---|---|
| 1 | 27.5 | 3.1 (4.6) | 0.03 | 23.5 | 4.5 | 19 | 0.06 | -0.42 | -0.38 |
| 2 | 6.7 | 20.3 (9.7) | 0.10 | 5.5 | 2.5 | 45 | -0.26 | -0.19 | -0.36 |
| 3 | 4.4 | 64.2 (8.7) | 0.17 | 2.0 | 1.0 | 50 | 0.28 | -0.21 | -0.22 |
| 4 | 8.4 | 23.4 (6.9) | 0.12 | 6.5 | 3.0 | 46 | 0.01 | **-0.52** | -0.41 |
| 5 | 5.5 | 44.0 (8.2) | 0.16 | 3.5 | 3.5 | 100 | -0.03 | **-0.48** | -0.41 |
| 6 | 3.1 | 78.1 (7.1) | 0.12 | 1.0 | 1.0 | 100 | -0.07 | 0.08 | 0.28 |
| 7 | 5.2 | 51.2 (9.5) | 0.19 | 4.5 | 2.5 | 56 | -0.10 | 0.08 | 0.11 |
| 8 | 5.5 | 85.9 (9.0) | 0.12 | 4.0 | 2.0 | 50 | -0.13 | -0.25 | -0.23 |
| 9 | 33.6 | 96.5 (4.1) | 0.03 | 30.5 | 13.0 | 43 | 0.09 | **0.92** | **0.67** |

## 5 Discussion

### 5.1 Cumulative melting surface and air temperatures

Previous studies have often used the cumulative melting surface (CMS) as an annual metric to describe the interannual variability of surface melt in Antarctica (e.g. Zwally and Fiegles, 1994; Picard and Fily, 2006; Trusel et al., 2012). For the Shackleton Ice Shelf, we find that interannual variations in the CMS strongly positively correlate ($r = 0.92$; $p < 0.01$) with the number of pattern 9 days each summer (Table 1), and that both of these have strong positive correlations with average summer (DJ) air temperatures across the shelf in RACMO2.3p3 (see Fig. 7b). These correlations support a significant role for near-surface temperature trends in driving interannual melt variability on the Shackleton Ice Shelf.

For the remaining patterns (1-—8), we observe no statistically significant correlations between the timing of their occurrence and the shelf-wide average air temperatures (Table 1). This observation suggests that these patterns are not driven by the larger-scale atmospheric circulation, but rather by differences in the local climate across the shelf. Furthermore, significant negative correlations between the CMS and the annual occurrences of patterns 4 and 5 (Table 1) show that in summers with less melt overall, the melt that does occur becomes more spatially fragmented across the shelf, further highlighting the role of local climate drivers.

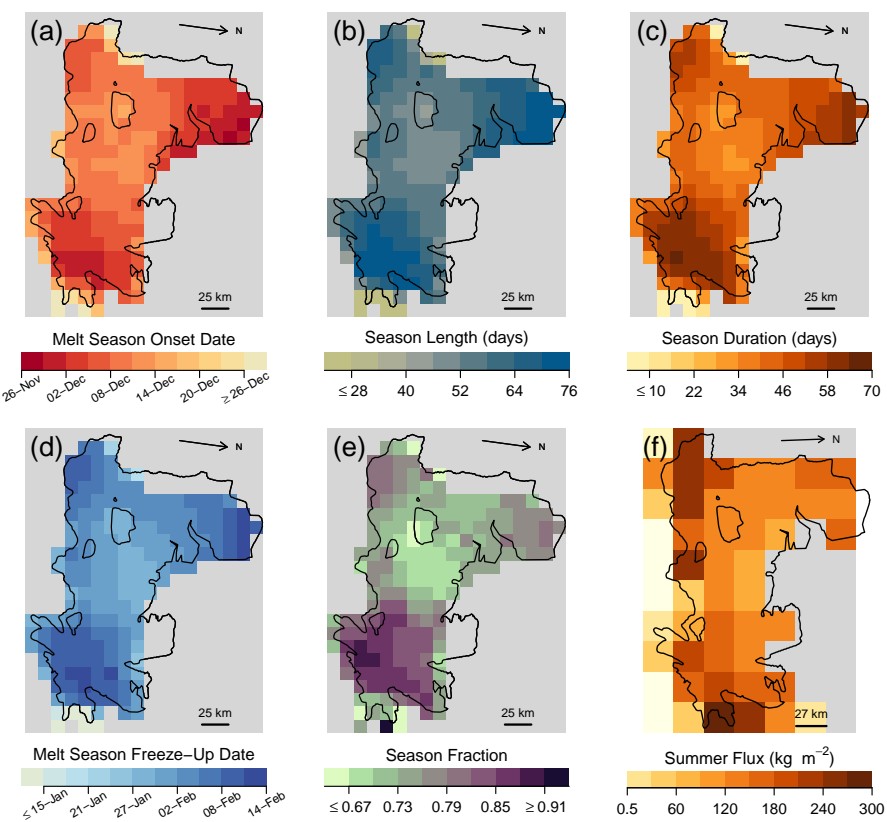

**Figure 6.** Average values across all melt seasons on a per-pixel basis, for (a) melt season onset date; (b) melt season length; (c) melt season duration; (d) melt season freeze-up date; (e) the fraction of the melt season experiencing melt, equivalent to plot (c) divided by plot (b); and (f) average cumulative summer (NDJF) melt flux from RACMO2.3p3 (van Dalum et al., 2021). Note that plots (a–e) include data for summers 2002/03–2010/11 and 2012/13–2020/21, whereas plot (f) includes data for 2002/03–2017/18; correlations between (c) and (f) discussed in Sect. 5.4 use contemporary dates.

## 5.2 Local controls on surface melt

Patterns 2–7 show that the occurrence of melt is not always uniform across the shelf, and suggest four approximate melt "zones", located in the south-east; to the south-west along the grounding line; in the centre of the shelf; and to the north. We suggest that the potential for heterogeneous melt behaviour in these four locations is because of their different geographic settings, and thus reveals the influence of local controls on surface melt.

In East Antarctica, strong katabatic winds are a persistent feature of the climate. Zones of convergent airflow channel and enhance the winds towards the Shackleton Ice Shelf (Parish and Bromwich, 2007), where they descend across the steep topography of the grounding line (Fig. 1; Fig. 7e). Despite being negatively buoyant (i.e. comparatively cooler than the surrounding air), on average, katabatic winds warm the underlying surface by disrupting the temperature inversion (Bromwich, 1989),

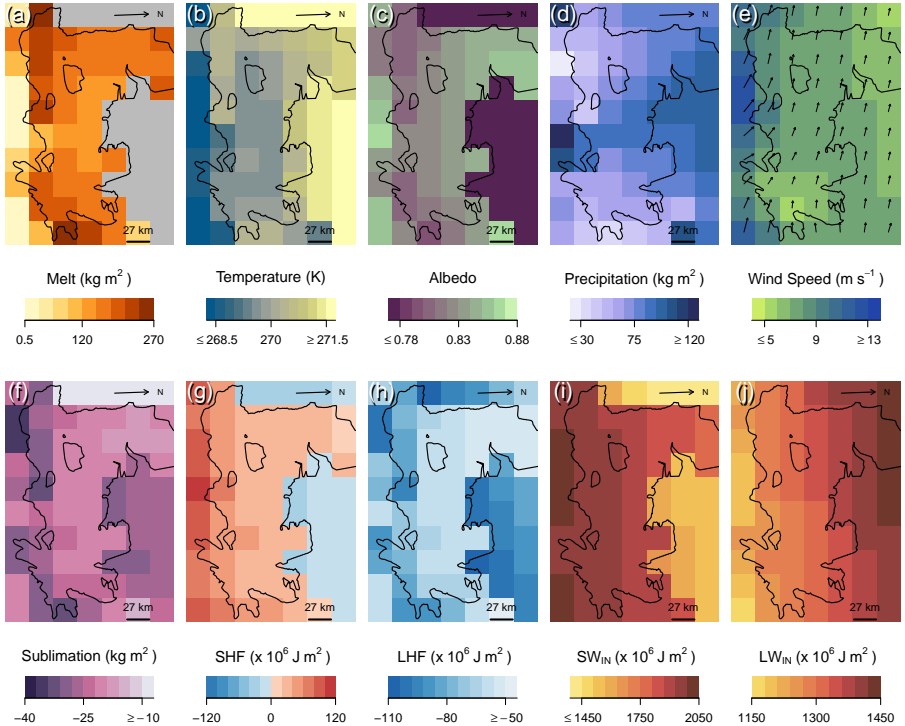

**Figure 7.** Average summer (DJ) values from RACMO2.3p3, for the Shackleton Ice Shelf (2002/03–2017/18). Plots (a), (d) and (f–j) show the average cumulative summer value; plots (b–c) and (e) are the average value across all summers. Plots (g–j) respectively show sensible heat flux (SHF), latent heat flux (LHF), incoming shortwave radiation (SW), and incoming longwave radiation (LW).

as well as scouring the surface and exposing areas of lower albedo that promote further melt through the snowmelt albedo feedback (Lenaerts et al., 2017a; Jakobs et al., 2019).

The role of katabatic winds in driving melt is supported by pattern 7, which shows melt occurring all along the grounding line. Patterns 4 and 5 also show that melt can occur along the grounding line in a small, isolated area to the south-west, whilst the shelf immediately further north remains dry. Such spatially restricted melt is indicative of a localised driver, and thus these

280 observations likely show katabatic wind-driven melt at the grounding line, potentially enhanced by the melt-albedo feedback as seen on the Roi Baudouin Ice Shelf (Lenaerts et al., 2017a). Average summer (DJ) values from RACMO2.3p3 show faster wind speeds along the grounding line (Fig. 7e), particularly to the south-west and over the Roscoe Glacier, which also have higher 2 m air temperatures (Fig. 7b) and increased sublimation losses (Fig. 7f) compared to their surroundings, consistent with localised katabatic conditions.

To the south-east of the shelf, melt is often observed when the rest of the shelf remains dry (i.e. pattern 4) and occurs extensively in all patterns except 1 and 2, suggesting a persistent, localised driver. Modelled wind speeds to the south-east are lower than to the south-west (Fig. 7e), potentially owing to the relatively coarse (27 km) resolution of RACMO2, which

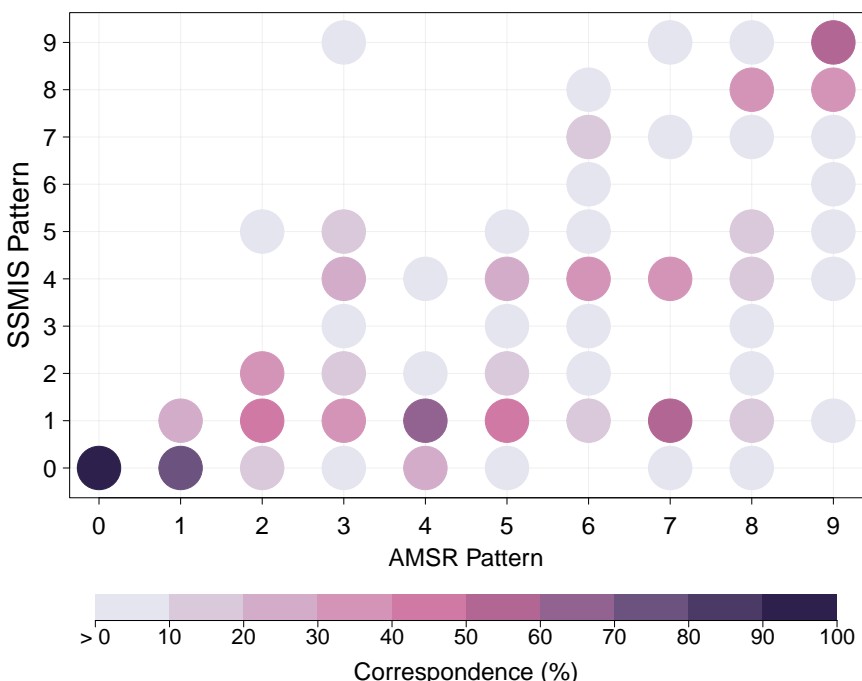

**Figure 8.** Correspondence of the daily spatial melt distribution for the AMSR and SSMIS sensors. Observations from both sensors are mapped against the nine melt patterns (Fig. 2). Colours indicate the percentage of days on which the pattern assigned to the AMSR data matches the pattern assigned to the SSMIS data. Pattern 0 indicates days outside of the melt season as defined by the respective sensor, and are thus not assigned a SOM pattern.

is known to underestimate wind speeds over steep topography (Lenaerts et al., 2012; van Wessem et al., 2018). However, a large expanse of exposed blue ice to the south-east of the shelf (Fig. 1a; Zheng and Zhou, 2020; Hui et al., 2014) shows that katabatic winds there are strong enough to scour the surface of fresh snow and keep the surface albedo low (Das et al., 2013; Lenaerts et al., 2017b). Also located to the south-east of the shelf is the low-albedo, ice-free Bunger Hills region (Colhoun and Adamson, 1989; Burton-Johnson et al., 2016), where strong katabatic winds have been observed descending down the Apfel Glacier towards the Shackleton Ice Shelf, able to raise winter air temperatures by up to 30 ℃ (Doran et al., 1996). Together, interactions between strong katabatic winds, large, low-albedo regions, and the snowmelt-albedo feedback can likely explain the south-east's increased susceptibility to melt: compared to the rest of the shelf, melt in the south-east is observed more consistently through the melt season (Fig. 6e), starts earlier (Fig. 6a), and ends later (Fig. 6d). However, further research is necessary to understand the interactions in greater detail, particularly given that supraglacial lakes are common in the south-east (Arthur et al., 2020b).

Pattern 2 shows that melt can occur to the north of the shelf in isolation from the grounding line, thus indicating that katabatic processes are not responsible. The propensity for pattern 2 to occur at the beginning and end of a melt season suggests the

influence of warmer summer air temperatures advancing across the shelf earlier and retreating later; the tendency for pattern 2 to retreat to pattern 1 on the following day (Fig. 4) shows that these are often short-lived melt events and could be driven by passing weather systems. On average, air temperatures over the north of the shelf are warmer than the rest of the shelf (Fig. 7b), and incoming longwave and shortwave radiation are consistent with heavier cloud cover along the calving front than at the grounding line (Fig. 7i–j). Cloud cover is an important control on the spatial variability of the surface energy balance in Antarctica (van den Broeke et al., 2006; van Wessem et al., 2014b; Scott et al., 2019), and is able to enhance melt by thermally blanketing the surface (Ghiz et al., 2021) and preventing refreezing (van Tricht et al., 2016).

In contrast to the three melt zones discussed above, patterns 3 and 6 show a clear melt-free region in the centre of the shelf, located between Masson Island and an area of fast ice that is near-permanent (Fraser et al., 2021). This central zone has a higher surface elevation (45–55 m above sea level; Fig. 1b) than its surroundings to the north and west (25–35 m) (Stephenson and Zwally, 1989; Howat et al., 2019), and cooler average summer temperatures (~ 0.5 K lower; Fig. 7b). It also has increased precipitation (Fig. 7d) and a very slightly higher average albedo (~ 1–2 %; Fig. 7c). Each of these factors can inhibit melt, and together can likely explain why summer melt usually begins later in the centre of the shelf than over the surrounding shelf (Fig. 6a) and freezes up earlier (Fig. 6d), as well as melting on a much smaller fraction of days in between (Fig. 6e). Furthermore, our observation that the centre of the shelf only melts when the full shelf melts (i.e. patterns 8 and 9) suggests this part of the shelf is only sensitive to more widespread drivers of melt (i.e. air temperatures), and that melt-feedback processes alone are not sufficient to sustain melt in the shelf centre.

## 5.3  Sensor comparisons

Few previous studies have investigated surface melt on the Shackleton Ice Shelf in detail, but there is some evidence of heterogeneous melt behaviour (see e.g. Fig. 3 of Liu et al., 2006; Fig. 2b of Tedesco and Monaghan, 2009; and Fig. 6 of Trusel et al., 2012). However, results can be sensor-dependent. For example, a longer melt season is observed in the north of the shelf with an active microwave sensor (ASCAT), but to the south-east with a passive sensor (AMSR-2) (see Fig. 2d of Zheng and Zhou, 2020). The frequency of the sensor also determines how many melt days are observed (see Fig. 3 & Fig. 6 of Leduc-Leballeur et al., 2020).

Whilst it can be tempting to try to establish which of the various datasets is 'correct', we advocate building on the approach of Leduc-Leballeur et al. (2020) and viewing different sensors as providing complementary rather than competing datasets. Differences between the sensors (e.g. sensor technology, frequency, spatial resolution, overpass time) can provide multiple perspectives on melt, and can be used to move beyond simple identification of melt occurrence and towards a better understanding of melt intensity and liquid water depth.

### 5.3.1  AMSR–SSMIS comparison

We compare our AMSR melt observations against those from the SSMIS sensors, and find that the two datasets only agree on the SOM melt pattern ~ 30 % of the time overall (Fig. 8). Because the assigned SOM patterns represent the observed melt

well for both datasets (Fig. S2 and S9–10), this low correspondence is mainly the result of three discrepancies in the sensors' observations.

Firstly, a third of all pattern 9 days in the AMSR dataset are mapped as pattern 8 days in the SSMIS observations. These two patterns are very similar, and the melt observations for each often only differ in the melt status of a couple of pixels. For many purposes, these patterns could therefore be considered equivalent between the sensors, though pattern 9 represents truer shelf-wide melt.

Secondly, the sensors disagree on when the melt season begins and ends each summer (i.e. the melt onset and freeze-up dates). Nearly a quarter of all days across the AMSR-defined melt seasons occur beyond the SSMIS-defined dates. Many of these days also have little melt in the AMSR observations (i.e. pattern 1), but, on average, there are 3.5 days each summer with extensive melt (mainly patterns 2 & 4) on days outside of the SSMIS-defined season. A clear diurnal melt cycle exists in East Antarctica (Picard and Fily, 2006), suggesting that the afternoon passes of the AMSR sensors are more likely to observe melt than the evening and morning passes of the SSMIS observations. Because we use a binary definition of melt, the effects of a diurnal cycle are likely to be more pronounced away from the height of the melt season, and could thus explain this discrepancy.

Thirdly, patterns 3 and 5–7 are rarely observed in the SSMIS dataset at all, and are only observed simultaneously in both datasets on 11 days over the entire study period. Differences in the spatial resolution of the sensors can become very important for melt observations in less homogeneous areas of an ice shelf, particularly if there are changes in elevation within the footprint (Johnson et al., 2020). The rarity of patterns 3, 6, and 7 in the SSMIS dataset is likely because of the sensor's coarser resolution as each of these three patterns show extensive melt along the steep grounding line. Likewise, pattern 7 in the AMSR data often corresponds to pattern 4 in the SSMIS data, and thus represents a reduction in melt along the grounding line between the two sensors. However, the grounding line does not remain melt-free in all SSMIS observations, indicating that sufficiently intense melt can compensate for higher elevation topography within the sensor's underlying footprint.

Together, the different resolutions and overpass times of the sensors can explain why the two datasets do not always agree, and suggest that discrepancies between different sensors can potentially provide additional information on the diurnal variability and relative intensity of melt. Future work can build on our SOM-based approach to help identify and understand such differences in further detail, and also extend comparisons to include other datasets used in the melt literature, such as ASCAT, SMOS, and Sentinel-1 (e.g. Bevan et al., 2018; Zhou et al., 2019; Johnson et al., 2020; Leduc-Leballeur et al., 2020; Banwell et al., 2021; Liang et al., 2021).

## 5.4 Melt days and melt fluxes

In this study, we use the concept of melt days, based on a binary definition of whether liquid water is observable at the shelf surface when the satellite passes overhead. At sufficiently large spatial scales, the number of melt days correlates well with meltwater production (Trusel et al., 2012). Visually inspecting maps of averaged AMSR melt season duration (Fig. 6c) and RACMO melt fluxes (Fig. 6f) shows a broad general agreement in their respective spatial variability, but correlations between the two are statistically insignificant regardless of the resampling approach used. This discrepancy is in part because of the coarser resolution of the RACMO dataset, and also because of differences along the grounding line.

Comparisons with QuikSCAT-derived melt fluxes suggest that RACMO overestimates melt along the grounding line by ~ 25–75 mm w.e. yr$^{-1}$ (see Fig. 12d of van Dalum et al., 2022), but melt observations from both passive and active microwave sensors can be greatly affected by the rock outcrops and blue ice found in and around the Bunger Hills (Zheng and Zhou, 2020). Because no in situ melt data exist for the Shackleton Ice Shelf, it is therefore not possible to verify the accuracy of flux estimates across the shelf, nor quantify the relationship between melt days and melt fluxes at this finer spatial scale.

Although we use melt days in the current work, our SOM-based approach could be adapted to incorporate melt fluxes in future work. In such a case, it is likely that using more than nine melt patterns would become beneficial because meltwater fluxes could help to further differentiate melt behaviour within each of the nine patterns we observe here and therefore help to identify the importance of different processes in driving melt. Furthermore, a SOM-based approach could also be adopted to facilitate comparisons between observational datasets and model output in more detail than is possible with annual or decadal averages.

## 5.5  Future melt

Modelling work has shown that melt rates on the Shackleton Ice Shelf could approach current-day values on the Larsen C Ice Shelf by 2100 (Trusel et al., 2015, their Fig. 3c). Meltwater runoff is also likely to greatly increase (Kittel et al., 2021, their Fig. 6), and potentially occur on up to 70 days each summer should temperatures reach 4 ℃ above pre-industrial values (Gilbert and Kittel, 2021). However, these models do not agree on whether the most intense melt fluxes on the shelf occur in the north (Trusel et al., 2015, their Fig. 4b) or south-east (Gilbert and Kittel, 2021, their Fig. 1). Continued satellite observations are necessary for ongoing evaluation of such models.

## 6  Conclusions

We use a self-organising map and daily passive microwave data from the AMSR-E and AMSR-2 sensors to identify nine representative patterns of surface melt on the Shackleton Ice Shelf over the past two decades (2002/03–2020/21).

The nine melt patterns show that the occurrence, extent and duration of surface melt are determined by both larger-scale temperature trends and local controls, specifically surface topography, albedo and winds. Our results support the importance of katabatic winds in driving melt on East Antarctic ice shelves, and suggest that the feedbacks between katabatic winds and surface albedo can initiate and prolong the melt season along the grounding line even when the rest of the shelf remains frozen. Nevertheless, strong correlations between the number of days of shelf-wide melt each summer, the cumulative melting surface (CMS), and the average summer (DJ) air temperature point to the importance of the larger-scale climate as a control on the interannual variability of surface melt on the Shackleton Ice Shelf.

Future work could use our approach to investigate the spatial and temporal variability of surface melt on other Antarctic ice shelves in greater detail than previously possible (see Code & Availability). Our approach could also be adapted to exploit publicly available satellite datasets to investigate surface melt across the cryosphere worldwide.

*Code and data availability.* The AMSR-E and AMSR-2 melt data will be made available online for the final paper. The R code used in this analysis can be accessed online at: github.com/polarSaunderson/ShackletonSOM; a doi link will be provided for the final paper.

RACMO2.3p3 model output (van Dalum et al., 2021) is publicly available online at: https://doi.org/10.5281/zenodo.5512077. LIMA (Bindschadler et al., 2008) and MEaSURES (Mouginot et al., 2017) are publicly available online and also accessible within Quantarctica (Matsuoka et al., 2021). REMA (Howat et al., 2019) is available online at https://www.pgc.umn.edu/data/rema/.

*Author contributions.* DS conceived of the study, performed the analysis, and wrote the manuscript, with guidance and input from all authors at each stage. GP processed the passive microwave satellite data. All authors contributed to discussions and commented on the manuscript.

*Competing interests.* The authors declare no competing interests.

*Acknowledgements.* We would like to thank the reviewers for their time and insightful comments, which have helped make the manuscript stronger. We also thank Christiaan van Dalum for providing the RACMO2.3p3 wind vectors used in Fig. 7e. DS was supported by the Monash Graduate Scholarship (MGS) and Monash International Tuition Scholarship (MITS). This research was carried out as part of the Australian Research Council (ARC) SRIEAS grant SR200100005, Securing Antarctica's Environmental Future. FM and RSJ were also respectively

supported by ARC Discovery Early Career Research Awards DE210101433 and DE210101923. GP was supported by ESA/AO/1-9570/18/I-DT - 4DAntarctica.

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
