# Peer review of "Surface melt on the Shackleton Ice Shelf, East Antarctica (2003–2021)"

_The Cryosphere, 2022_

## Referee Comment (RC1)

**Surface melt on the Shackleton Ice Shelf, East Antarctica (2003-2021)**

Dominic Saunderson, Andrew Mackintosh, Felicity McCormack, Richard S. Jones, Ghislain Picard.

Jennifer Arthur (Referee)
jennifer.arthur@durham.ac.uk

**General comments:**

This manuscript investigates the spatiotemporal variability in summer surface melt on the Shackleton Ice Shelf over the past two decades (2002/3 – 2020/21). The authors use a machine learning approach (a self-organising map) and passive microwave AMSRE data to identify nine representative patterns of surface melt on the ice shelf over this period. Finally, they use these patterns to demonstrate the importance of surface air temperatures, katabatic winds and surface albedo in controlling summer melt.

Surface melt is an important component of ice-sheet surface mass balance, leading to surface meltwater accumulating in depressions on ice-shelf surfaces. This has been linked to the process of meltwater-driven hydrofracture, which can trigger rapid ice-shelf collapse. Quantifying surface melt extent and duration and understanding the controls on this variability is therefore important for evaluating future melt projections. Melt metrics from satellite-derived observations have typically been used to quantify surface melt. However, few studies have quantified the extent and occurrence of surface melt and its interannual variability on East Antarctic ice shelves.

This manuscript builds upon previous work by providing the first assessment of interannual variability in summer surface melt on the Shackleton Ice Shelf, potentially one of the most vulnerable ice shelves in East Antarctica to future meltwater-driven collapse. The novel approach of a self-organising map is used to evaluate this, the first time this has been done in Antarctica. It also provides useful insight into the local controls on the occurrence of surface melt, building upon recent work linking localised katabatic wind controls on seasonal supraglacial lake evolution on this ice shelf.

Therefore, it is my view that the findings are of broad interest to the cryospheric community and represent a promising step forward for studying the spatial and temporal variability of surface melt and its controls, especially in the context of ice-shelf surface hydrology and dynamics. I look forward to seeing further development of this method and its applications on other Antarctic ice shelves.

In general, this is a well-written manuscript and most of my comments are relatively minor. Once the authors address these, I can therefore recommend that this manuscript is suitable for publication in *The Cryosphere*.

**Specific comments:**

Line 29: I think it is worth highlighting here how intense surface melt can precondition ice shelves for collapse by depleting their firn air content. Think about adding an additional sentence or two here about the specific role played by anomalously intense surface melt preceding the break-up of Larsen B (e.g. van den Broeke, 2005).

Line 34: 'melt events' → specify timescale, i.e. over several days. Similarly on Line 35: 'longer records' → specify multiannual.

Line 36: Specify melt metrics which are typically used (melt onset/freeze-up dates, melt season length, total number of melt days).

Line 61: I suggest citing Miles et al. (2020) here, who recorded acceleration of Denman Glacier since 1972, driven by grounding line retreat, ice tongue thinning and unpinning (see References below).

Line 71: Define SSM/I and SSMIS acronyms.

Line 84: I would perhaps clarify here how you define dry snow recursively?

Line 95: It isn't clear to me what you mean by erroneously missing melt events – do you mean you want to exclude any melting that could be occurring on the sea ice?

Line 119: I would refer to Section S1 in the Supplement here, with the useful description for understanding the SOM output.

Line 165: I would change the order in which the different patterns are discussed, and displayed in figures 2 and 5 – it is slightly confusing to start with patterns 8 and 9 (though I can see why you have done this as they are the most prevalent). Consider re-ordering sequentially.

Line 176: Specify what you define as the melt season (i.e. I assume November to February?).

Line 192 (Figure 3): I think it would be useful to show calendar dates on this figure (i.e. merge Fig. S6 with this one). Would it be possible to add a second x axis to show this? The colours of SOM Patterns 1 and 7 are also very similar, consider changing one.

Line 217: I think Figure S8 is interesting and should be brought into the main manuscript. While Fig. 5 is useful for demonstrating the large interannual variability, particularly in the two 'extreme' end members i.e. Patterns 9 and 1, I think Fig. S8 provides useful context for interpreting the temporal variability in melt patterns across the shelf.

Line 229: Could you add Cumulative Melt Surface into the Table caption to remind the reader, given that the other acronyms are defined here.

Line 261: Add a brief summary of RACMO2.3p3 in Section 2, as it is not currently mentioned in the description of datasets. Is it forced with ERA5? And reference van Dalum et al. (2022).

Line 244 (Figure 6): Consider in Panels (a) and (d) changing scale units from 'Days since 1st Nov' to Date, I think this would make it easier to see spatial variability through the melt season.

Line 342: Could you quantify how much RACMO overestimates melt along the grounding line? It looks like ~25-75 mm w.e yr$^{-1}$ from van Dalum et al. (2022)?

**Technical/minor corrections:**

Hyphenate 'sea-level rise' (e.g. Line 15 – check throughout).

Line 54: I suggest replacing 'runs for' with 'extends for'.

Line 55: Replace 'lay' with 'lie'.

Line 78: 19 GHz (space).

Line 81: Hyphenate 'horizontally-polarised'.

Section 5.3: Capitalise 'figure' throughout.

Line 356:  → 'do not'. Same on Line 328.

**References**

Van den Broeke (2005) Strong surface melting preceded collapse of Antarctic Peninsula ice shelf. *Atmospheric Science*, 32, 12. Doi:10.1029/2005GL023247.

Miles, B. et al. (2020) Recent acceleration of Denman Glacier (1972–2017), East Antarctica, driven by grounding line retreat and changes in ice tongue configuration. *The Cryosphere*, 15, 663–676. https://doi.org/10.5194/tc-15-663-2021.

---

## Referee Comment (RC2)

**Surface melt on the Shackleton Ice Shelf, East Antarctica (2003–2021)**

Dominic Saunderson, Andrew Mackintosh, Felicity McCormack, Richard S. Jones, and Ghislain Picard

Review by Ian Willis for TC

This is a very interesting, novel, clearly structured and well written paper that was a pleasure to read. Thank you to the authors for spending the time to hone the material so that it was logically presented, the writing was succinct, clear, precise and accurate, and it contained very few grammatical errors as far as I could tell. I wish all papers I was asked to review were of this calibre.

The science is also robust, with one or two small exceptions the discussion does not overinterpret things, and the conclusions logically follow on from the results presented.

The paper uses the machine learning approach of self-organised maps (SOMs), which has been used in other branches of the environmental sciences (notably climatology) but has not previously been used, as far as the authors (or I) know, in the field of glaciology in this way. The method is not dissimilar to the EOF approach but, as the authors state on lines 124-6, has advantages. The method is applied to Advanced Microwave Scanning Radiometer data and used to investigate melt patterns across the Shackleton Ice Shelf over an 18-year period. The nine 'modal' patterns (Fig 2) are a useful way of reducing the complex spatial and temporal dataset into a series of manageable and interpretable outputs. The sequencing of these patterns throughout the melt seasons (Figs 3 & 4) are instructive in identifying how melt patterns typically evolve through the summer. This, together with the interannual variability in this sequencing, is helpful in terms of thinking about the processes driving melt across the ice shelf. The sensitivity analysis to model parameters (Sect 3.3 & Figs S4-S5) is valuable.

The paper also works with the original binary melt files to map patterns of 'seasonal metrics' that have been mapped on other ice shelves by others, notably melt onset, season length, season duration, melt season freeze up date and the fraction of the melt season experiencing melt (Fig 6).

Finally, the paper goes some way towards explaining the spatial and temporal melt patterns in terms of climate model output (esp. air temperature, albedo) and previous work (esp. that discussing katabatic winds).

I believe the paper makes a valuable contribution to the growing literature investigating melt variability on ice shelves and their drivers. This work is important since increased melting on ice shelves increases their vulnerability to break up and collapse.

The paper provides access to the SOM code (written in R-studio) and gives examples of how the approach could be altered and /or used in other similar contexts.

Like the writing, the Figures are produced to a high standard and are quickly interpretable and relevant.

I therefore find no major problems with the paper and suggest only that the authors consider the following moderate and small suggestions / questions and small editorial / typographical issues that I outline in turn below.

**Moderate suggestions / questions**

L175. You provide us with the average values for how much of the ice shelf is experiencing melt on days which conform to the 9 patterns. Would it also be useful to provide a measure of variability (e.g. S.D.) in this %?

L229-230. You describe a situation for 2008/09 here but what about 2007/8 where pattern 9 is also v low but pattern 3 not particularly high? Pattern 5 is high in 2007/8, which also has a dry centre. Is this significant and worth mentioning?

L234. This section 4.5 has done a nice job at looking at interannual variability in the patterns of melt. What are the correlations between the annual patterns? Could you calculate the correlation coefficients between the data in Fig 5 to quantify the extent to which the patterns co-vary directly, inversely, or not at all?

L243-4. I think the suggestion that melt patterns 1-8 are not climatically driven needs some qualification. The patterns themselves presumably are climatically driven, in the sense that when there's high melt in one place and not another, the climate will be different between the places? It is the number of days of the year with a particular pattern that is uncorrelated with average summer air temps at an interannual timescale. Are they correlated with some other aspect of air temp / climate?

L258. You say the "role of katabatic winds in driving melt is supported by patterns 3–7". Is this true? Is it not just pattern 7 which supports it, with melt all along the grounding line and very little elsewhere? In the other patterns, how do you explain the occurrence of high melt in the N close to the edge but not at the GL? Is this not due to other mechanisms? How do intra and interannual variability of patterns showing absence of melt in centre match with variability of precip. as Fig 7d suggests months / years with high pptn. (assuming it is snow) would be associated with patterns 4 and 5 in particular. Perhaps you address these point in the discussion but of not it would be worth doing so.

L264-271. Given the focus on winds explaining patterns here (and in fact in this whole section 5.2), it seems odd that the wind vectors from RACMO are not shown in Fig 7. Could this be done? I think it'd be instructive. Could you also see if there are differences in the wind field during the different melt patterns?

L363. Could "…such as…" be changed to "…specifically…" and could you list all the local controls that you show / infer are relevant? Presumably wind field is relevant here? I suggested above that you might show the RACMO wind field vectors in Fig 7. You refer to

albedo here and show that in Fig 7. Could you also show a map of surface topography in the discussion section?

**Small suggestions / questions**

L1-2. add the intervening processes of ice shelf disintegration & grounded ice acceleration if room.

L2-3. As above, this sentence could be made clearer with a little more explanation if the word count allows.

L3. Clarify it is previous studies of surface melt on ice shelves here I assume?

L4. Can you explain better / give an example of a 'regional melt metric' as it's not obvious here [I know this is explained in the main body of text but not until Section 2.5 on lines 100-105, and it'd be useful if it were made more apparent in the Abstract].

L9-11. It's not obvious how you're able to identify a significant role for 'air tempertaures' and 'local factors' Can you state that you're using RACMO output and a DEM to do this?

L30. As above, clarify it is previous studies of surface melt on ice shelves here I assume?

L36. As above, can you explain better / give an example of a 'regional melt metric' as it's still not obvious here.

L41. Add Dell et al 2021 here? They show maximum melt extents and persistence for Roi Baudouin Ice Shelf, East Antarctica. Dell, R.L., Banwell, A.F., Willis, I.C., Arnold, N.S., Halberstadt, A.R.W., Chudley, T.R. and Pritchard, H.D., 2021. Supervised classification of slush and ponded water on Antarctic ice shelves using Landsat 8 imagery. Journal of Glaciology, 68, 401-414.

L45. "… and thus identify the influence of local controls on the occurrence of surface melt" As written so far, it's not clear precisely what you mean by this and how exactly you'll identify such controls. From what you said previously it looks like you're talking about albedo and air temperatures? Do you compare the melt patterns and variations with patterns and variations in albedo and air temperature? Useful to explain this a bit more clearly here.

L46. This sentence is rather lazily written and vague. Can you more precisely explain this?

L49. "…in relation to the local controls on surface melt" See comments above. We still don't have a sense of precisely what this means.

Fig 1. A nice map but check the legend against what's displayed and add more to the heading. Perhaps it's just my computer screen but I see brown as well as orange lines depicting the shelf boundary but just orange in the legend. I assume the thin line depicting the grounding line is consumed beneath the orange/brown line for most of the shelf but is

there for the eastern part. State in the heading where the grounding line data come from. Also, I don't see Mask Pixels on the map. And what are the large black linear features towards the edges of the two lobes of the Denman Glacier?

L82. "…annually and spatially adaptive threshold" Can this be explained a little more?

L84. "…dry snow is defined recursively" Expanding on this a little would be useful too.

L95. It doesn't look like 25 km has been removed from the front of the Denman Glacier in Fig 1.

L179-180. As this point is not specific to patterns 9 and 8 should this generic point be made outside these sentences about 9 and 8?

Fig 4 is v interesting. One thing that strikes me is that pattern 8 is 20-30% likely to progress to pattern 9 but not the reverse. What does this tell us about melt processes on the ice shelf? Also, melt pattern 9 is approx. equally likely (0-10%) to become any other pattern (except pattern 1). What does this tell us about melt processes? Perhaps you discuss these points later in the text so apologies if that is the case. But if not, it may be useful to address them?

L204-5. How to explain pattern 2 (melt at N edge) being most common early and late season but pattern 4 (which doesn't have much melt towards N edge) being most common overall? What is control on "switching on and off" melt at N edge?

Table 1. This is quite instructive. ~1/3 of the time there's virtually no melt; ~1/3 of the time there's virtually 100% melt; ~1/3 of the time there's one of 7 other patterns, ranging in freq. between c. 3% and 8% of the time.

Fig 6. Vs Fig 7. Is there a reason why summer is defined as NDJF for Fig 6 but just DJ for Fig 7?

L272-3. Yes, see my earlier comment (against L258) on this.

L284-5. Yes, I agree the higher elevation and the fact that the centre of the ice shelf experiences more pptn (Fig 7d) is likely important here.

L332-3. Banwell et al 2021 could be added to this ref list. I believe they used SMMR, SSMI as well as ASCAT.

L339. How do you do this correlation given the different spatial resolutions?

L342. You say RACMO overestimates melt cf. QuikSCAT Is this also true here in your study with the AMSR data? I don't think you tell us whether RACMO melt is over or underestimated.

L348-350. "…because meltwater fluxes would help to differentiate between melt behaviour considered equivalent here and could help identify how important different processes are for melt." I don't follow this. Can it be rewritten to make clearer?

L364. I suggest change "show" to "suggest" here. I think you need to be more circumspect. I'm not convinced you've really proved beyond doubt the role of feedbacks as you state here.

**Small editorial / typographical issues**

L27 Suggest change 'force' to 'stress' here.

L36. '…describe quantitatively…'

L39. Suggest "Studies using melt metrics that have focussed…"

L156. Should strictly be "data have….and are thus…" [i.e. plural]

L170-182. These three short paragraphs (inc. 2 x 1 sentence paragraphs) could be merged. Check entire document for this.

Fig 1 Heading. Suggest say "…the shelf boundary adapted from the MEaSURES dataset…"

L61-2. Suggest say "Understanding the response of the Denman Glacier and wider Shackleton system to climate variability and change is therefore an important area of research".

L73. Suggest change to "pre-processed in the same way as the AMSR datasets"

Fig 3 Heading "…data are plotted…"

L223. 'observe' to be consistent (present tense) with L218.

L228. Suggest "…summer (Fig. 5; Fig. S8) provides a snapshot"

L286. Suggest change to "summer melt usually begins later in the centre of the shelf than over the surrounding shelf" which is more grammatically correct.

L287. Suggest "…much smaller fraction…"

L319-20. Suggest delete words "occurring" and "between them".

L328. "…do not always agree…"

L345. Strictly "data exist" [plural]

L347. Suggest "…approach could be adapted…"

L353. "…on the Larsen C…"

L356 "… do not…"

L369. "…work could use…"

L370 "…approach could also…"

---

## Author Comment (AC1)

**Author Response to Reviewer 1 Comments on tc-2022-94**

Reviewer 1: Jennifer Arthur

**General Comments**

We would like to thank Jennifer for her time reviewing our manuscript and for her helpful suggestions; addressing these will definitely improve our work and make the manuscript stronger. We appreciate Jennifer's overview and understanding of our work, and her acknowledgement of its interest to the cryospheric research community. We are also grateful for her comments on the manuscript being well written, and her recommendation that, after addressing her comments, the manuscript is suitable for publication in The Cryosphere.

Below we address each of Jennifer's comments in turn: Jennifer's comments are shown in *red italics*, followed by our Author Response (AR) in blue, and our suggested amendments.

**Specific Comments**

**Line 29:** *I think it is worth highlighting here how intense surface melt can precondition ice shelves for collapse by depleting their firn air content. Think about adding an additional sentence or two here about the specific role played by anomalously intense surface melt preceding the break-up of Larsen B (e.g. van den Broeke, 2005).*
**AR:** We will amend the text on lines 23-26 to better highlight the anomalous melt intensity and the role of melt in reducing firn air content:

"One of the most notable examples of ice shelf collapse occurred in 2002, when the Larsen B Ice Shelf disintegrated in six weeks (Rack and Rott, 2004), following intense surface melt (Sergienko & MacAyeal, 2005; van den Broeke, 2005) and extensive meltwater ponding across the shelf surface (Leeson et al., 2020). Such ponding occurs when the firn layer becomes saturated, depleting the firn air content of the shelf (Kuipers Munneke et al., 2014; Holland et al., 2011; Luckman et al., 2014), and allowing liquid water to collect in depressions on the ice shelf surface (Arthur et al., 2020a)."

**Line 34:** *'melt events' → specify timescale, i.e. over several days. Similarly on Line 35: 'longer records' → specify multiannual*
**AR:** We will amend the text to specify these timescales:

"…examining individual melt events (i.e. over several days) (Zou et al., 2019; Ghiz et al., 2021) or a few melt seasons (Elvidge et al., 2020; Turton et al., 2020), though some longer, multiannual records do exist (Jakobs et al., 2020)."

**Line 36:** *Specify melt metrics which are typically used (melt onset/freeze-up dates, melt season length, total number of melt days).*

**AR:** A similar comment was made by the second reviewer. We will amend the text to specify typical melt metrics:

"Secondly, previous studies describe quantitatively the occurrence and extent of melt in Antarctica using a series of melt metrics calculated from satellite observations. Typical metrics include the melt onset and freeze-up dates each summer, the total number of melt days, and the cumulative melting surface (e.g. Zwally and Fiegles, 1994; Torinesi et al., 2003). These metrics are often reported at a regional (e.g. Antarctic Peninsula, Wilkes Land) or continental scale, and usually show…"

**_Line 61_**_: I suggest citing Miles et al. (2020) here, who recorded acceleration of Denman Glacier since 1972, driven by grounding line retreat, ice tongue thinning and unpinning (see References below)._
**AR:** We will make a slight change to the text to include this reference:

"The Denman Glacier is estimated to have lost ~ 190 Gt of ice since 1979 (~ 0.5 mm of sea-level rise; Rignot et al., 2019), and has accelerated over both its grounded and floating portions (1972—2017; Miles et al., 2021), with its grounding line having retreated nearly 5.5 km between 1996 and 2018 (Brancato et al., 2020; Konrad et al., 2018)."

**_Line 71:_** _Define SSM/I and SSMIS acronyms._
**AR:** We will add these definitions:

"… observations from the Special Sensor Microwave Imager (SSM/I) and the Special Sensor Microwave Imager Sounder (SSMIS) sensors: F13…"

**_Line 84:_** _I would perhaps clarify here how you define dry snow recursively?_
**AR:** This comment was also made by the second reviewer. We will expand our explanation and amend lines 82—85 to make it clearer:

"… algorithm used in Picard and Fily (2006). This algorithm uses a threshold approach to detect melt, with the threshold calculated for each pixel individually and redefined each summer. The threshold is calculated as the sum of the mean and 2.5 times the standard deviation of $T_B$ observations for dry snow each year (1st April—31st March). Dry snow is defined recursively, iteratively removing any observations identified as wet snow and recalculating the melt threshold using only the remaining observations, until no further observations need to be removed; one or two iterations are sufficient to reach convergence. A full explanation can be found in Torinesi et al. (2003)."

**_Line 95:_** _It isn't clear to me what you mean by erroneously missing melt events – do you mean you want to exclude any melting that could be occurring on the sea ice?_
**AR:** The salt content of sea ice increases the emissivity of the surface, resulting in a higher $T_B$ value even when the surface remains dry. If pixels affected by these higher dry $T_B$ values subsequently melt, the increase in $T_B$ from the dry surface to the wet surface (i.e. our definition of melt) may not be large enough to consistently trigger our melt algorithm, and therefore genuine melt events could be missed. Furthermore, a polynya is often observed off the western coast of the shelf, meaning that often the dry $T_B$ values are actually the result of a varying combination of dry snow, sea ice, and open ocean. Because our algorithm is designed only for a wet vs. dry surface distinction, it is not suitable for

use on pixels which could also be influenced by these additional surface "types". We will amend the text and simplify our explanation to better reflect this situation:

"Secondly, we exclude the westernmost edge of the shelf, which is often bordered by a polynya (Nihashi and Ohshima, 2015). Manual inspection of the underlying $T_B$ data indicates that the pixels along the western edge may have been contaminated by the inclusion of sea ice and open ocean in the sensor footprint, and are therefore not suitable for use with our algorithm, which is only designed to differentiate between wet and dry snow. The final shelf mask…"

We will add this reference to the bibliography:

Nihashi, S. and K.I. Ohshima.: Circumpolar Mapping of Antarctic Coastal Polynyas and Landfast Sea Ice: Relationship and Variability, Journal of Climate, 28, 3650—3670, https://doi.org/10.1175/JCLI-D-14-00369.1, 2015.

***Line 119:*** *I would refer to Section S1 in the Supplement here, with the useful description for understanding the SOM output.*
**AR:** We will add a reference at the end of this explanation on line 123, pointing the reader to Section S1 for further details:

"… repeatedly fed into the algorithm until it converges to a solution (see Sect. 1 in the Supplement for further explanation). The self-organisation…"

***Line 165:*** *I would change the order in which the different patterns are discussed, and displayed in figures 2 and 5 – it is slightly confusing to start with patterns 8 and 9 (though I can see why you have done this as they are the most prevalent). Consider re-ordering sequentially.*
**AR:** Our preference is to leave the structure of the results section unchanged because we feel that grouping the patterns in this way better highlights the main takeaways of these results. We also previously tried to present the patterns in order (i.e. 1—9) but received feedback that it was harder for the reader to follow.

***Line 176:*** *Specify what you define as the melt season (i.e. I assume November to February?).*
**AR:** When we refer to the melt season in the results and Figure 3, we are referring to the period between the dates of melt onset and freeze-up each summer, meaning that the actual dates differ each summer. The difference in these dates between summers is why we have used the melt context in Figure 3, as it allows us to refer to the relative timing of the patterns within a summer. We will amend the text to clarify this here:

"Both of these patterns can occur throughout the melt season (i.e. any time between the melt season onset and freeze-up dates), but are most prevalent…"

***Line 192 (Figure 3):*** *I think it would be useful to show calendar dates on this figure (i.e. merge Fig. S6 with this one). Would it be possible to add a second x axis to show this? The colours of SOM Patterns 1 and 7 are also very similar, consider changing one.*
**AR:** We do not think that it would be possible to merge the data from Figures 3 and S6 in a clearly presentable way because the different x-axes do not correspond exactly and thus the data are grouped differently in these two figures (see previous comment re: line 176). We agree that calendar dates (i.e. Fig. S6) are more immediately intuitive, but they also

present a slightly different story, particularly at the beginning and end of the melt season. For example, using the melt context (Fig. 3), we are able to see that any of patterns 1, 2, 4, 7, 8 and 9 can be observed right at the start of the melt season, whereas Fig. S6 shows that only patterns 1 and 2 occur early on. The differences between these two plots are interesting in themselves, but we felt that it made more sense to discuss the timing of the patterns in a relative way and thus used the melt context (Figure 3) in the main text, but included a reference to Fig. S6 in the caption to allow comparisons.

Regarding the colours of these plots, we used a colourblind safe palette but agree that some of the colours can be difficult to distinguish. We will therefore add hatching to help differentiate between similar colours (e.g. 1 and 7; 2, 3 and 6).

**_Line 217:_** _I think Figure S8 is interesting and should be brought into the main manuscript. While Fig. 5 is useful for demonstrating the large interannual variability, particularly in the two 'extreme' end members i.e. Patterns 9 and 1, I think Fig. S8 provides useful context for interpreting the temporal variability in melt patterns across the shelf._
**AR:** We agree that Fig. S8 is interesting and are happy to move it into the main manuscript as suggested in place of Fig. 5. Both plots use the same underlying data but tell slightly different stories (as this comment acknowledges), hence why we included both. We will adjust references to these figures in the text accordingly.

**_Line 229:_** _Could you add Cumulative Melt Surface into the Table caption to remind the reader, given that the other acronyms are defined here._
**AR:** We will add this to the Table caption as requested.

**_Line 261:_** _Add a brief summary of RACMO2.3p3 in Section 2, as it is not currently mentioned in the description of datasets. Is it forced with ERA5? And reference van Dalum et al. (2022)._
**AR:** We agree that this would be a helpful addition and will add a small subsection (Sect. 2.6) to introduce the data.

"2.6 Climate variables

We use monthly climate variables from the RACMO2.3p3 regional climate model (van Dalum et al., 2021; 2022), which is the latest version of the RACMO model that has been used extensively in Antarctica (e.g. Lenaerts et al., 2012; van Wessem et al., 2014; 2018). RACMO2.3p3 includes updates to the albedo scheme and multilayer firn module, as well as allowing subsurface penetration of shortwave radiation, which can be important for melt in Antarctica (Liston and Winther, 2005; Liston et al., 1999). The model is forced at its lateral boundaries by ERA5 reanalysis data (Hersbach et al., 2020)."

**AR:** We will add the following references to the bibliography:

Hersbach, H., Bell, B., Berrisford, P., Hirahara, S., Horányi, A., Muñoz-Sabater, J., Nicolas, J., Peubey, C., Radu, R., Schepers, D., Simmons, A., Soci, C., Abdalla, S., Abellan, X., Balsamo, G., Bechtold, P., Biavati, G., Bidlot, J., Bonavita, M., Chiara, G. D., Dahlgren, P., Dee, D., Diamantakis, M., Dragani, R., Flemming, J., Forbes, R., Fuentes, M., Geer, A., Haimberger, L., Healy, S., Hogan, R. J., Hólm, E., Janisková, M., Keeley, S., Laloyaux, P., Lopez, P., Lupu, C., Radnoti, G., Rosnay, P. de, Rozum, I., Vamborg, F., Villaume, S., and Thépaut, J.-N.: The ERA5 global reanalysis, Quarterly Journal of the Royal Meteorological Society, 146, 1999–2049, https://doi.org/10.1002/qj.3803, 2020.

Lenaerts, J. T. M., van den Broeke, M. R., Scarchilli, C., and Agosta, C.: Impact of model resolution on simulated wind, drifting snow and surface mass balance in Terre Adélie, East Antarctica, Journal of Glaciology, 58, 821–829, https://doi.org/10.3189/2012JoG12J020, 2012.

Liston, G. E. and Winther, J.-G.: Antarctic Surface and Subsurface Snow and Ice Melt Fluxes, J. Climate, 18, 1469–1481, https://doi.org/10.1175/JCLI3344.1, 2005.

Liston, G. E., Winther, J.-G., Bruland, O., Elvehøy, H., and Sand, K.: Below-surface ice melt on the coastal Antarctic ice sheet, Journal of Glaciology, 45, 273–285, https://doi.org/10.3189/S0022143000001775, 1999.

van Wessem, J. M., Reijmer, C. H., Morlighem, M., Mouginot, J., Rignot, E., Medley, B., Joughin, I., Wouters, B., Depoorter, M. A., Bamber, J. L., Lenaerts, J. T. M., van de Berg, W. J., van den Broeke, M. R., and van Meijgaard, E.: Improved representation of East Antarctic surface mass balance in a regional atmospheric climate model, Journal of Glaciology, 60, 761–770, https://doi.org/10.3189/2014JoG14J051, 2014.

van Wessem, J. M., van de Berg, W. J., Noël, B. P. Y., van Meijgaard, E., Amory, C., Birnbaum, G., Jakobs, C. L., Krüger, K., Lenaerts, J. T. M., Lhermitte, S., Ligtenberg, S. R. M., Medley, B., Reijmer, C. H., Tricht, K. van, Trusel, L. D., van Ulft, L. H., Wouters, B., Wuite, J., and van den Broeke, M. R.: Modelling the climate and surface mass balance of polar ice sheets using RACMO2 – Part 2: Antarctica (1979–2016), The Cryosphere, 12, 1479–1498, https://doi.org/10.5194/tc-12-1479-2018, 2018.

***Line 244 (Figure 6):*** *Consider in Panels (a) and (d) changing scale units from 'Days since 1st Nov' to Date, I think this would make it easier to see spatial variability through the melt season.*
**AR:** We agree that calendar dates are easier to read and will update the units of the figure as suggested.

***Line 324:*** *Could you quantify how much RACMO overestimates melt along the grounding line? It looks like ~25-75 mm w.e yr-1 from van Dalum et al. (2022)?*
**AR:** We will add an approximate value for this overestimation based on Figure 12d of van Dalum et al. (2022):

"Comparisons with QuikSCAT-derived melt fluxes suggest that RACMO overestimates melt along the grounding line by $\sim$ 25—75 mm w.e. yr$^{-1}$ (see Fig. 12d of van Dalum et al., 2022), but melt observations…"

**Technical/minor corrections**

*Hyphenate 'sea-level rise' (e.g. **Line 15** – check throughout).*
***Line 54:*** *I suggest replacing 'runs for' with 'extends for.'*
***Line 55:*** *Replace 'lay' with 'lie.'*
***Line 78:*** *19 GHz (space).*
***Line 81:*** *Hyphenate 'horizontally-polarised.'*
***Section 5.3:*** *Capitalise 'figure' throughout.*
***Line 356:*** *don't → 'do not.' Same on Line 328.*
**AR:** We will correct these in the manuscript - thank you for noticing them!

---

## Author Comment (AC2)

**Author Response to Reviewer 2 Comments on tc-2022-94**

Reviewer 2: Ian Willis

**General Comments**

We would like to thank Ian for his time reviewing our manuscript and for his insights on how to improve the manuscript further. We really appreciate Ian's supportive comments, particularly regarding the novelty of the approach, the robustness of the science, and his acknowledgement of the effort that was put into writing this manuscript. We are also grateful that Ian recognises our work as a valuable contribution in this growing field.

Below we address each of Ian's comments in turn: Ian's comments are shown in *red italics*, followed by our Author Response (AR) in blue, and our suggested amendments.

**Moderate suggestions / questions**

**L175:** *You provide us with the average values for how much of the ice shelf is experiencing melt on days which conform to the 9 patterns. Would it also be useful to provide a measure of variability (e.g. S.D.) in this %?*
**AR:** We will add the variability to Table 1 and amend the text to include this information in the examples given:

"… on average, ~ 86 % (SD = 9.0 %) of the shelf experiences melt on pattern 8 days, compared to ~ 96.5 % (SD = 4.1 %) on pattern 9 days (Table 1; Fig. S2d)."

**L229-230.** *You describe a situation for 2008/09 here but what about 2007/8 where pattern 9 is also v low but pattern 3 not particularly high? Pattern 5 is high in 2007/8, which also has a dry centre. Is this significant and worth mentioning?*
**AR:** We agree that 2007/08 is an interesting summer that shares many similarities with 2008/09. We are currently investigating these summers for a different paper exploring the temporal variability of surface melt in greater detail and better understanding drivers of interannual variability.

We will amend the text to include the 2007/08 summer as an additional example:

"For example, in 2007/08 and 2008/09, minimums in the occurrence of pattern 9 days and maximums in the occurrence of patterns 5 and 3, respectively, show that the shelf centre remained unusually dry for most of these melt seasons. In contrast, in 2010/11, …"

**L234.** *This section 4.5 has done a nice job at looking at interannual variability in the patterns of melt. What are the correlations between the annual patterns? Could you calculate the correlation coefficients between the data in Fig 5 to quantify the extent to which the patterns co-vary directly, inversely, or not at all?*
**AR:** We discuss this very briefly at the end of Sect. 4.1 (line 181) and Sect. 4.3 (line 208), where we write that patterns 8 and 9 are negatively correlated (r = -0.49), and that

pattern 5 correlates positively with pattern 8 (r = 0.49) but negatively with pattern 9 (r = -0.61). All other correlations are statistically insignificant at p < 0.05, although patterns 1 & 9 are negatively correlated (r = -0.46) at p = 0.055.

We will amend the text on line 218 to explicitly state the lack of correlations, and also add a table (Table S1) to the Supplement to show the correlation coefficients between patterns.

"We do not observe any statistically significant trends in the occurrence of any patterns on an interannual basis (Table 1; Fig 5), nor any statistically significant covariation in the patterns (Table S1) other than in patterns 5, 8 and 9 as discussed above. Rather…"

We will also amend the text on lines 181 and 208 to refer to this new table:

L181: "Interannually, the occurrence of patterns 8 and 9 are negatively correlated (r = -0.49; p = 0.04; Table S1), and only…"

L208: "Annually, the occurrence of pattern 5 is positively correlated with that of pattern 8 (r = 0.49; p = 0.04) and negatively correlated with that of pattern 9 (r = -0.61; p = 0.01; Table S1)."

***L243-4.*** *I think the suggestion that melt patterns 1-8 are not climatically driven needs some qualification. The patterns themselves presumably are climatically driven, in the sense that when there's high melt in one place and not another, the climate will be different between the places? It is the number of days of the year with a particular pattern that is uncorrelated with average summer air temps at an interannual timescale. Are they correlated with some other aspect of air temp / climate?*

**AR:** We accept that our use of "climatically-driven" lacked precision here. Rather than "climatically-driven", we wanted to suggest that patterns 1—8 are probably not driven by the larger-scale atmospheric circulation, which would more likely bring shelf-wide conditions. We will amend the text on lines 243—246 to clarify this:

"For the remaining patterns (1—8), we observe no statistically significant correlations between the timing of their occurrence and the shelf-wide average air temperatures (Table 1). This observation suggests that these patterns are not driven by the larger-scale atmospheric circulation, but rather by differences in the local climate across the shelf. Furthermore, significant negative correlations between the CMS and the annual occurrences of patterns 4 and 5 (Table 1) show that in summers with less melt overall, the melt that does occur becomes more spatially fragmented across the shelf, further highlighting the role of local climate drivers."

***L258.*** *You say the "role of katabatic winds in driving melt is supported by patterns 3–7." Is this true? Is it not just pattern 7 which supports it, with melt all along the grounding line and very little elsewhere? In the other patterns, how do you explain the occurrence of high melt in the N close to the edge but not at the GL? Is this not due to other mechanisms? How do intra and interannual variability of patterns showing absence of melt in centre match with variability of precip. as Fig 7d suggests months / years with high pptn. (assuming it is snow) would be associated with patterns 4 and 5 in particular. Perhaps you address these point in the discussion but of not it would be worth doing so.*

**AR:** We agree that pattern 7 definitely supports the importance of katabatic winds but we also think that patterns 4 and 5 do as well. Although patterns 4 and 5 observe melt in the north as well as at the grounding line, the isolation of the melt to the south-west suggests a localised melt driver there, likely katabatic winds funnelled down to the shelf by the topography of the Roscoe Glacier, and the interaction of the katabatic winds with melt and surface albedo that has been discussed elsewhere in East Antarctica by Lenaerts et al. (2017). Pattern 5 may therefore actually be a combination of the drivers in pattern 2 (i.e. northerly melt potentially driven by clouds/passing weather systems) and those of pattern 4 (i.e. extensive melt in the south-east and isolated melt to the south-west, driven by katabatic winds and the melt-albedo feedback).

We do however agree that the link to katabatic winds for patterns 3 and 6 was more speculative and will therefore remove reference to these patterns from this subsection. We will amend our wording (starting on line 258) to better reflect these interpretations:

"The role of katabatic winds in driving melt is supported by pattern 7, which shows melt occurring all along the grounding line. Patterns 4 and 5 also show that melt can occur along the grounding line in a small, isolated area to the south-west, whilst the shelf immediately further north remains dry. Such spatially restricted melt is indicative of a localised driver, and thus these observations likely show katabatic wind-driven melt at the grounding line, potentially enhanced by the melt-albedo feedback as seen on the Roi Baudouin Ice Shelf (Lenaerts et al., 2017). Average summer values…"

Regarding the variability of precipitation, this is something that we are currently investigating for a separate paper specifically concerned with better understanding the interannual variability of melt on the Shackleton Ice Shelf (as noted in our comment above re: Line 229). The relationships between the relevant variables are not immediately straightforward (i.e. the specific timing, location and sequencing of melt and precipitation can be important), plus there are multiple variables involved such as air temperature and radiative plus turbulent fluxes that can all interact, and we also expect extreme events to be important. Delving into these aspects would require a lot more work and we would prefer to focus on the spatial variability of melt across the shelf in the current paper, and therefore consider that this work would be out of scope here.

**_L264-271._** _Given the focus on winds explaining patterns here (and in fact in this whole section 5.2), it seems odd that the wind vectors from RACMO are not shown in Fig 7. Could this be done? I think it'd be instructive. Could you also see if there are differences in the wind field during the different melt patterns?_

**AR:** Originally, we did not include the RACMO wind vectors because they were not publicly available, but we have since contacted the corresponding author (Christiaan van Dalum), who has kindly shared these datasets for use in the paper. We will add an extra subplot to Fig. 7 showing the average summer (DJ) wind speed and direction, and rearrange the figure to better accommodate 10 subplots:

[Figure]

We will correct references to these subplots throughout the text, and will also make some amendments to the text to refer to this additional wind plot and to better reflect the results it shows:

Line 253: "… enhance the winds towards the Shackleton Ice Shelf (Parish and Bromwich, 2007), where they descend across the steep topography of the grounding line (Fig. 1; Fig. 7e). Despite being…"

Line 261: "Average summer (DJ) values from RACMO2.3p3 show faster wind speeds along the grounding line (Fig. 7e), particularly to the south-west and over the Roscoe Glacier, which also have higher 2 m air temperatures (Fig. 7b) and increased sublimation losses (Fig. 7f) compared to their surroundings, consistent with localised katabatic conditions."

Line 265: "To the south-east of the shelf, melt is often observed when the rest of the shelf remains dry (i.e. pattern 4) and occurs extensively in all patterns except 1 and 2, suggesting a persistent, localised driver. Modelled wind speeds to the south-east are much lower than to the south-west (Fig. 7e), potentially owing to the relatively coarse (27 km) resolution of RACMO2, which is known to underestimate wind speeds over steep topography (Lenaerts et al., 2012; van Wessem et al., 2018). However, a large expanse of blue ice to the south-east of the shelf (Fig. 1; Zheng and Zhou, 2020; Hui et al., 2014) shows that katabatic winds there are strong enough to scour the surface of fresh snow and keep the surface albedo low (Das et al., 2013; Lenaerts et al., 2017). Also located to the south-east of the shelf is the low-albedo, ice-free Bunger Hills region (Colhoun and Adamson, 1989; Burton-Johnson et al., 2016), where strong katabatic winds have been observed descending down the Apfel Glacier towards the Shackleton Ice Shelf, able to raise winter air temperatures by up to 30 ºC (Doran et al., 1996). Together, interactions between strong katabatic winds, large, low-albedo regions, and the snowmelt-albedo

feedback can likely explain the south-east's increased susceptibility to melt: compared to the rest of the shelf, melt in the south-east is observed more consistently through the melt season (Fig. 6e), starts earlier (Fig. 6a) and ends later (Fig. 6d). However, further research is necessary to understand the interactions in greater detail, particularly given that supraglacial lakes are common in the south-east (Arthur et al., 2020b)."

We will add the following references:

Lenaerts, J. T. M., van den Broeke, M. R., Scarchilli, C., and Agosta, C.: Impact of model resolution on simulated wind, drifting snow and surface mass balance in Terre Adélie, East Antarctica, Journal of Glaciology, 58, 821–829, https://doi.org/10.3189/2012JoG12J020, 2012.

van Wessem, J. M., van de Berg, W. J., Noël, B. P. Y., van Meijgaard, E., Amory, C., Birnbaum, G., Jakobs, C. L., Krüger, K., Lenaerts, J. T. M., Lhermitte, S., Ligtenberg, S. R. M., Medley, B., Reijmer, C. H., Tricht, K. van, Trusel, L. D., van Ulft, L. H., Wouters, B., Wuite, J., and van den Broeke, M. R.: Modelling the climate and surface mass balance of polar ice sheets using RACMO2 – Part 2: Antarctica (1979–2016), The Cryosphere, 12, 1479–1498, https://doi.org/10.5194/tc-12-1479-2018, 2018.

Regarding investigating differences in the winds (or any other climatic variables) between melt patterns, the RACMO data we have used is monthly-averaged, and therefore unsuitable to investigate the daily melt patterns in this much detail. We are hopeful that we can investigate this idea further in upcoming work.

***L363.*** *Could "…such as…" be changed to "…specifically…" and could you list all the local controls that you show / infer are relevant? Presumably wind field is relevant here? I suggested above that you might show the RACMO wind field vectors in Fig 7. You refer to albedo here and show that in Fig 7. Could you also show a map of surface topography in the discussion section?*
**AR:** We will add a map of the shelf's surface topography as an inset to Fig. 1, including elevation contours across the ice shelf, and will refer to this on line 282 when we discuss topography:

"… higher surface elevation (45–55 m above sea level; Fig. 1) than its surroundings to the north and west (25–35 m) (Stephenson and Zwally, 1989; Howat et al., 2019), and cooler…"

We will also amend the text on line 363 to list the relevant controls as requested:

"… and local controls, specifically surface topography, albedo, and winds."

**Small suggestions / questions**

***L1-2.*** *add the intervening processes of ice shelf disintegration & grounded ice acceleration if room.*
***L2-3.*** *As above, this sentence could be made clearer with a little more explanation if the word count allows.*
**AR:** We will rewrite the opening lines of the abstract to address these comments:

"Melt on the surface of Antarctic ice shelves can potentially lead to their disintegration, accelerating the flow of grounded ice to the ocean and raising global sea levels. However, the current understanding of the processes driving surface melt is incomplete, increasing

uncertainty in predictions of ice shelf stability and thus of Antarctica's contribution to sea level rise."

**L3.** *Clarify it is previous studies of surface melt on ice shelves here I assume?*
**AR:** Whilst melt in Antarctica is predominantly observed on ice shelves, many of the previous studies also include non-shelf areas in their analysis, particularly in the regional metrics (e.g. Zwally and Fiegles, 1994; Trusel et al., 2012), and some have even explicitly looked at how far inland or up to what elevation melt has been observed (e.g. Tedesco et al., 2007). We can however clarify that we are referring to previous studies of surface melt in Antarctica:

"Previous studies of surface melt in Antarctica..."

**L4.** *Can you explain better / give an example of a 'regional melt metric' as it's not obvious here [I know this is explained in the main body of text but not until Section 2.5 on lines 100-105, and it'd be useful if it were made more apparent in the Abstract].*
**AR:** We will amend the text to include an example:

"... or used metrics such as the annual number of melt days to quantify spatiotemporal variability in satellite observations of surface melt."

**L9-11.** *It's not obvious how you're able to identify a significant role for 'air tempertaures' and 'local factors' Can you state that you're using RACMO output and a DEM to do this?*
**AR:** We will clarify this sentence to include the use of RACMO and the REMA DEM:

"Combined with output from the RACMO2.3p3 regional climate model and surface topography from the REMA digital elevation model, our results point to a..."

**AR:** In order to make the above changes to the abstract text and still adhere to the word limit (250 word), we will make minor adjustments to the abstract text. In full, the abstract will therefore read:

"Melt on the surface of Antarctic ice shelves can potentially lead to their disintegration, accelerating the flow of grounded ice to the ocean and raising global sea levels. However, the current understanding of the processes driving surface melt is incomplete, increasing uncertainty in predictions of ice shelf stability and thus of Antarctica's contribution to sea level rise. Previous studies of surface melt in Antarctica have usually focused on either a process-level understanding of melt through energy-balance investigations, or used metrics such as the annual number of melt days to quantify spatiotemporal variability in satellite observations of surface melt. Here, we help bridge the gap between work at these two scales. Using daily passive microwave observations from the AMSR-E and AMSR-2 sensors, and the machine learning approach of a self-organising map, we identify nine representative spatial distributions ("patterns") of surface melt on the Shackleton Ice Shelf, East Antarctica from 2002/03—2020/21. Combined with output from the RACMO2.3p3 regional climate model and surface topography from the REMA digital elevation model, our results point to a significant role for surface air temperatures in controlling the interannual variability of summer melt, and also reveal the influence of localised controls on melt. In particular, prolonged melt along the grounding line shows the importance of katabatic winds and surface albedo. Our approach highlights the necessity of understanding both local and large-scale controls on surface melt, and

demonstrates that self-organising maps can be used to investigate the variability of surface melt on Antarctic ice shelves."

**L30.** *As above, clarify it is previous studies of surface melt on ice shelves here I assume?*
**AR:** As above (L3), we will clarify that we are discussing melt in Antarctica:

"Broadly speaking, previous research has investigated the occurrence of surface melt in Antarctica in one of two ways."

**L36.** *As above, can you explain better / give an example of a 'regional melt metric' as it's still not obvious here.*
**AR:** A similar comment was made by the first reviewer. We will amend the text to specify typical melt metrics:

"Secondly, previous studies describe quantitatively the occurrence and extent of melt in Antarctica using a series of melt metrics calculated from satellite observations. Typical metrics include the melt onset and freeze-up dates each summer, the total number of melt days, and the cumulative melting surface (e.g. Zwally and Fiegles, 1994; Torinesi et al., 2003). These metrics are often reported at a regional (e.g. Antarctic Peninsula, Wilkes Land) or continental scale, and usually show…"

**L41.** *Add Dell et al 2021 here? They show maximum melt extents and persistence for Roi Baudouin Ice Shelf, East Antarctica. Dell, R.L., Banwell, A.F., Willis, I.C., Arnold, N.S., Halberstadt, A.R.W., Chudley, T.R. and Pritchard, H.D., 2021. Supervised classification of slush and ponded water on Antarctic ice shelves using Landsat 8 imagery. Journal of Glaciology, 68, 401-414.*
**AR:** We will add this reference and update our bibliography:

"… with only a few in East Antarctica (Zhou et al., 2019; Zheng & Zhou, 2020; Dell et al., 2021)."

**L45.** *"… and thus identify the influence of local controls on the occurrence of surface melt" As written so far, it's not clear precisely what you mean by this and how exactly you'll identify such controls. From what you said previously it looks like you're talking about albedo and air temperatures? Do you compare the melt patterns and variations with patterns and variations in albedo and air temperature? Useful to explain this a bit more clearly here.*
**AR:** We will amend this sentence (starting on line 44) to clarify what we mean by local controls and how we identify them:

"We use a machine learning approach to assess the inter- and intra-annual variability in the location of satellite-observed surface melt, and compare our results with surface topography and patterns of climate variables such as surface air temperatures and albedo to identify the influence of localised controls on the occurrence of surface melt."

**L46.** *This sentence is rather lazily written and vague. Can you more precisely explain this?*
**AR:** Our intention here was to place our research in context with the two preceding paragraphs, where we made a broad distinction between SEB and metric-based remote sensing studies. We will amend this sentence to be more precise:

"...of surface melt. Our approach therefore goes beyond the metrics used in previous remote sensing studies and begins to bridge the gap between spatiotemporal descriptions of melt variability derived from satellite observations, and the process-level understanding of melt from SEB studies that are more detailed but limited in time and space."

***L49.*** *"...in relation to the local controls on surface melt" See comments above. We still don't have a sense of precisely what this means*
**AR:** We will amend this sentence to:

"In Sect. 4, we present the results, which we then discuss in Sect. 5; particular attention is given to understanding the results in relation to the local geographic setting (e.g. surface topography, albedo, and winds) and its role in controlling surface melt."

***Fig 1.*** *A nice map but check the legend against what's displayed and add more to the heading. Perhaps it's just my computer screen but I see brown as well as orange lines depicting the shelf boundary but just orange in the legend. I assume the thin line depicting the grounding line is consumed beneath the orange/brown line for most of the shelf but is there for the eastern part. State in the heading where the grounding line data come from. Also, I don't see Mask Pixels on the map. And what are the large black linear features towards the edges of the two lobes of the Denman Glacier?*
**AR:** We will darken the line colours to improve the clarity of the mask pixels in this figure. We will also correct the colours of the grounding line and shelf boundaries, and amend the caption to make it clear that both the grounding line and shelf boundary come from MEaSURES.

Regarding the large black linear features, these are areas of open ocean that are visible because the LIMA imagery is from 2000-2003 (published 2008) and the MEaSURES boundary is newer (published 2017). Neither of these regions are in the shelf mask (see comment re: Line 95), and thus this does not affect our results. We will add the dates of these datasets to the caption to explain this difference.

***L82.*** *"...annually and spatially adaptive threshold" Can this be explained a little more?*
***L84.*** *"...dry snow is defined recursively" Expanding on this a little would be useful too.*
**AR:** A similar comment was made by the first reviewer. We will amend the text (starting on line 82) to make this explanation clearer:

"... algorithm used in Picard and Fily (2006). This algorithm uses a threshold approach to detect melt, with the threshold calculated for each pixel individually and redefined each summer. The threshold is calculated as the sum of the mean and 2.5 times the standard deviation of $T_B$ observations for dry snow each year (1st April—31st March). Dry snow is defined recursively, iteratively removing any observations identified as wet snow and recalculating the melt threshold using only the remaining observations, until no further observations need to be removed; one or two iterations are sufficient to reach convergence. A full explanation can be found in Torinesi et al. (2003)."

***L95.*** *It doesn't look like 25 km has been removed from the front of the Denman Glacier in Fig 1.*

**AR:** We will improve the clarity of the lines in Figure 1 (see our comment re: Fig 1 above) so that it is clear we have removed the front 25 km of the Denman Glacier from the melt grid used in our analysis, but not excluded it from this figure.

**L179-180.** *As this point is not specific to patterns 9 and 8 should this generic point be made outside these sentences about 9 and 8?*
**AR:** We agree that this sentence is more general than just patterns 8 and 9, but it does not fit easily in any of the subsections in the results. We suggest that we can move this point to the caption of Fig. 4:

"Figure 4. Day-to-day progressions between the nine melt patterns. The colours indicate how frequently a pattern on the x-axis develops into a pattern on the y-axis the following day, highlighting the strong tendency for any pattern to persist until at least the next day. White squares indicate that the progression is never observed."

**Fig 4** *is v interesting. One thing that strikes me is that pattern 8 is 20-30% likely to progress to pattern 9 but not the reverse. What does this tell us about melt processes on the ice shelf? Also, melt pattern 9 is approx. equally likely (0-10%) to become any other pattern (except pattern 1). What does this tell us about melt processes? Perhaps you discuss these points later in the text so apologies if that is the case. But if not, it may be useful to address them?*
**AR:** These are interesting observations! As noted in other comments, understanding the melt processes and drivers in greater detail is a subject of current research and we will look to investigate the transitions further as part of that work. However, in this study, we are hesitant to overinterpret what the transitions can tell us about melt processes as we have only monthly resolution climate datasets, and also because it is difficult to establish statistical significance given how infrequent these patterns are in absolute numbers (e.g. 8→9 occurs only 19 times and 9→8 only 25 times over our 18-year melt record). These absolute numbers are also very similar, which is not reflected in Fig. 4 because of how much more persistent pattern 9 is overall, and thus we are cautious to make comparisons in this way between different patterns in Fig. 4.

We also note that because > 90 % of pattern 9 days are followed by another pattern 9 day, transitions from pattern 9 to the remaining patterns (i.e. 2—8) are visualised equally in Fig. 4 (i.e. coloured in the 0—10 % interval). However, there is some variability in the frequency of these transitions, with a higher frequency of transitions from 9→8 (4.6 %) and 9→6 (2.2 %) than any of the others (each < 1.5 %).

**L204-5.** *How to explain pattern 2 (melt at N edge) being most common early and late season but pattern 4 (which doesn't have much melt towards N edge) being most common overall? What is control on "switching on and off" melt at N edge?*
**AR:** We are also very intrigued by melt at the northern edge. We hypothesise that melt there could be driven largely by its more northerly latitude and protrusion into the ocean, which would expose it to passing weather systems and warmer air temperatures that may not reach across more of the shelf. During the middle of the melt season, we suspect that melt will also arise elsewhere on the shelf (e.g. from katabatic winds, lower albedo, shelf-wide increases in air temperature), and thus melt will no longer be restricted to the north (i.e. it's no longer pattern 2).

We will make a slight amendment in the text on lines 273-275 to make this idea clearer:

"The propensity for pattern 2 to occur at the beginning and end of a melt season suggests the influence of warmer summer air temperatures advancing across the shelf earlier and retreating later; the tendency for pattern 2 to retreat to pattern 1 on the following day (Fig. 4) shows that these are often short-lived melt events and could be driven by passing weather systems. On average, air temperatures…"

***Table 1.*** *This is quite instructive. ~1/3 of the time there's virtually no melt; ~1/3 of the time there's virtually 100% melt; ~1/3 of the time there's one of 7 other patterns, ranging in freq. between c. 3% and 8% of the time.*
**AR:** We agree that this is an interesting breakdown of melt on the Shackleton Ice Shelf and are intrigued to see whether these approximate-thirds apply to other shelves, or hold over the coming years. These proportions could have serious implications for the shelf in terms of firn recovery time, the likelihood of meltwater ponds, and susceptibility to hydrofracture + collapse.

***Fig 6. Vs Fig 7.*** *Is there a reason why summer is defined as NDJF for Fig 6 but just DJ for Fig 7?*
**AR:** The majority of melt occurs in December and January, and there is comparatively very little melt in November and/or February for many summers. Because we are using monthly-averaged variables from RACMO, we therefore decided to use only the DJ values to give a more representative climatology of the variables relating to melt each summer (i.e. Fig. 7b—7j). Following this, it made sense to also only display the DJ melt in Fig. 7a.

Conversely, in Fig. 6f, we are showing all melt from NDJF so that it matches Fig. 6a—6e, which include all satellite-observed melt and therefore span across NDJF. Comparing Fig. 6f and Fig. 7a, we can see that there is very little difference between the two, highlighting that most of the melt (at least in terms of melt fluxes, as modelled by RACMO) does indeed occur in December and January.

***L272-3.*** *Yes, see my earlier comment (against L258) on this.*
**AR:** We will amend the text slightly here (see above comments re: Line 204-5).

***L284-5.*** *Yes, I agree the higher elevation and the fact that the centre of the ice shelf experiences more pptn (Fig 7d) is likely important here.*
**AR:** We agree and feel that the threshold nature of melt means even small differences in such factors can have important consequences.

***L332-3.*** *Banwell et al 2021 could be added to this ref list. I believe they used SMMR, SSMI as well as ASCAT.*
**AR:** We will add this reference.

"… such as ASCAT, SMOS, and Sentinel-1 (e.g. Bevan et al., 2018; Zhou et al., 2019; Johnson et al., 2020; Leduc-Leballeur et al., 2020; Liang et al., 2021; Banwell et al., 2021)."

***L339.*** *How do you do this correlation given the different spatial resolutions?*
**AR:** We resampled the datasets to match each other's spatial resolutions, and tried both nearest neighbour and bilinear interpolation methods. In no circumstances were the correlations statistically significant. We will clarify this in the text:

"... but correlations between the datasets were statistically insignificant regardless of the resampling approach used. This discrepancy..."

**_L342._** _You say RACMO overestimates melt cf. QuikSCAT Is this also true here in your study with the AMSR data? I don't think you tell us whether RACMO melt is over or underestimated._

**AR:** We are unable to make a direct comparison between the AMSR data, which is only able to provide binary melt/no-melt information (line 335), and RACMO fluxes. This is why we visually compare maps of the AMSR melt days and RACMO fluxes (Fig. 6c & Fig. 6f; line 338) and attempt to correlate the two (line 339-340).

As for QuikSCAT, our comment on line 342 about QuikSCAT-RACMO differences is based purely on the figure in van Dalum et al. (2022) and we therefore do not have the QuikSCAT data to make direct comparisons with the AMSR data ourselves.

After a comment from the first reviewer, we will make a slight change on line 342 to estimate the amount that RACMO overestimates melt compared to QuikSCAT:

"Comparisons with QuikSCAT-derived melt fluxes suggest that RACMO overestimates melt along the grounding line by $\sim$ 25—75 mm w.e. yr$^{-1}$ (see Fig. 12d of van Dalum et al., 2022), but melt observations..."

**_L348-350._** _"...because meltwater fluxes would help to differentiate between melt behaviour considered equivalent here and could help identify how important different processes are for melt." I don't follow this. Can it be rewritten to make clearer?_

**AR:** We will rewrite this sentence to make this point clearer:

"...because meltwater fluxes could help to further differentiate melt behaviour within each of the nine patterns we observe here and therefore help to identify the importance of different processes in driving melt."

**_L364._** _I suggest change "show" to "suggest" here. I think you need to be more circumspect. I'm not convinced you've really proved beyond doubt the role of feedbacks as you state here._

**AR:** We agree with this comment and will amend our language:

"... on East Antarctic ice shelves, and suggest that the feedbacks between katabatic winds and surface albedo..."

**Small editorial / typographical issues**

**_L27_** _Suggest change 'force' to 'stress' here._
**_L36._** _'...describe quantitatively...'_
**_L39._** _Suggest "Studies using melt metrics that have focussed..."_
**_L156._** _Should strictly be "data have....and are thus..." [i.e. plural]_
**_L170-182._** _These three short paragraphs (inc. 2 x 1 sentence paragraphs) could be merged. Check entire document for this._
**_Fig 1 Heading._** _Suggest say "...the shelf boundary adapted from the MEaSURES dataset..."_
**_L61-2._** _Suggest say "Understanding the response of the Denman Glacier and wider Shackleton system to climate variability and change is therefore an important area of research."_

*__L73.__ Suggest change to "pre-processed in the same way as the AMSR datasets"*
*__Fig 3 Heading__ "...data are plotted..."*
*__L223.__ 'observe' to be consistent (present tense) with L218.*
*__L228.__ Suggest "...summer (Fig. 5; Fig. S8) provides a snapshot"*
*__L286.__ Suggest change to "summer melt usually begins later in the centre of the shelf than over the surrounding shelf" which is more grammatically correct.*
*__L287.__ Suggest "...much smaller fraction..."*
*__L319-20.__ Suggest delete words "occurring" and "between them."*
*__L328.__ "...do not always agree..."*
*__L345.__ Strictly "data exist" [plural]*
*__L347.__ Suggest "...approach could be adapted..."*
*__L353.__ "...on the Larsen C..."*
*__L356__ "... do not..."*
*__L369.__ "...work could use..."*
*__L370__ "...approach could also..."*

__AR:__ We will correct all of these small issues in the manuscript as suggested. Thank you for noticing these!

---

## Author Response (AR1)

**Author Response**

We would like to thank the editor for his time looking at our author's responses to the reviewers' comments. We have now amended our manuscript in line with their suggestions. Please find attached the updated manuscript and a file tracking the changes, as well as an updated supplement.